

# Comparison of six approaches to predicting droplet activation of surface active aerosol – Part 2: strong surfactants

Sampo Vepsäläinen[1], Silvia M. Calderón[2], and Nønne L. Prisle[3]

[1]Nano and Molecular Systems Research Unit, University of Oulu, P.O. Box 3000, FI-90014, Oulu, Finland
[2]Finnish Meteorological Institute, P.O. Box 1627, FI-70211, Kuopio, Finland
[3]Center for Atmospheric Research, University of Oulu, P.O. Box 4500, FI-90014, Oulu, Finland

**Correspondence:** Nønne L.Prisle (nonne.prisle@oulu.fi)

**Abstract.** Surfactants have been a focus of investigation in atmospheric sciences for decades due to their ability to modify the water uptake and cloud formation potential of aerosols. Surfactants adsorb to the air–solution interface and can decrease the surface tension, while in microscopic aqueous droplets simultaneously depleting the droplet bulk. While this mechanism is now broadly accepted, the representation in atmospheric and cloud droplet models is still not well constrained. We compare the predictions of five bulk–surface partitioning models and a general bulk solution model documented in the literature to represent aerosol surface activity in Köhler calculations of cloud droplet activation. The models are applied to a suite of common aerosol particle systems, consisting of strong surfactants (sodium myristate or myristic acid) and sodium chloride in a wide range of relative mixing ratios. The partitioning models predict comparable critical droplet properties at small surfactant mass fractions, but differences between the model predictions for identical particles increase significantly with the surfactant mass fraction in the particles. For the same particles and simulation conditions, the partitioning models also predict significantly different surface compositions and surface tensions for growing droplets along the Köhler curves. The inter-model variation is furthermore different for these particles comprising strongly surface active organics, than for moderately surface active atmospheric aerosol components. Our results show that experimental validation across a range of atmospherically relevant aerosol compositions, surface active properties, and droplet states is necessary before a given model can be generally applied in atmospheric predictions.

## 1 Introduction

The global climate is affected by atmospheric aerosols both directly through interaction with solar radiation and indirectly through their ability to serve as cloud condensation nuclei (CCN). The indirect effect from cloud–aerosol radiation interactions still remains a large source of uncertainty to global radiative forcing estimates (IPCC, 2013, 2021). Surface active species (surfactants) are commonly found in atmospheric aerosols (e.g., Gérard et al., 2016; Petters and Petters, 2016; Nozière et al., 2017; Kroflič et al., 2018; Gérard et al., 2019). The presence of surfactants in aqueous droplets reduces the surface tension from



that of pure water and simultaneously causes mass transfer from the droplet bulk to the surface (e.g., Shulman et al., 1996; Li et al., 1998; Sorjamaa et al., 2004; Prisle et al., 2008, 2010; Nozière et al., 2014; Bzdek et al., 2020), through a process called
bulk–surface *partitioning*. In atmospheric aerosols, surfactants potentially affect the critical point of cloud droplet activation, but a clear consensus has not yet been reached on the magnitude and specific dynamics of the effect (e.g., Hänel, 1976; Shulman et al., 1996; Facchini et al., 1999; Facchini et al., 2000; Li et al., 1998; Sorjamaa et al., 2004; Prisle et al., 2008, 2010, 2011; Topping, 2010; Raatikainen and Laaksonen, 2011; Ruehl and Wilson, 2014; Nozière et al., 2014; Ruehl et al., 2016; Petters and Petters, 2016; Ovadnevaite et al., 2017; Malila and Prisle, 2018; Lin et al., 2018; Prisle et al., 2019; Davies et al., 2019;
Lowe et al., 2019; Lin et al., 2020; Bzdek et al., 2020; Prisle, 2021; Vepsäläinen et al., 2022).

Surface tension measurements are commonly performed for macroscopic solutions and have only recently been successfully performed for microscopic droplets (i.e., with diameters in the micrometer range or smaller) and for only a few selected droplet systems. Morris et al. (2015) performed measurements with NaCl, malonic, and glutaric acids, Bzdek et al. (2016) with NaCl and glutaric acid, and Bzdek et al. (2020) with Triton X-100. In microscopic droplets, the partitioning of the surfactant mass
between the droplet bulk and surface has been predicted to significantly alter the droplet bulk composition, due to the finite total amount of surfactant contained in such droplets, and therefore affect composition-dependent properties such as the surface tension (e.g., Prisle et al., 2010; Lin et al., 2018, 2020). The relation between surface tension and surface composition is typically unknown (Prisle et al., 2012; Werner et al., 2018), and surface tension is expressed in terms of the solution bulk composition. In a macroscopic solution with a total composition identical to that of a microscopic droplet, the bulk-phase contains
a sufficiently large amount of surfactant that bulk–surface partitioning has a negligible effect on the solution composition and a relation to the surface tension is easily established as the bulk and total compositions are practically identical. For microscopic droplets, the composition change due to bulk–surface partitioning can be modeled using *bulk–surface partitioning models*. The models calculate the partitioning between the droplet bulk and surface to correct the bulk composition of the droplets to account for the material partitioned to the droplet surface. This allows the use of macroscopic solution properties for microscopic
droplets. Recently, Bzdek et al. (2020) provided the first experimental evidence of the influence of bulk-phase depletion due to bulk–surface partitioning in microscopic finite sized microscopic droplets (7-9 $\mu$m radius) by experimentally observing the surface tension at the air/solution interface of droplets containing Triton X-100 suspended in air. The droplet surface tension was found to be higher than that of the macroscopic counterpart, indicating a change in the droplet bulk composition due to bulk–surface partitioning.

Several approaches have been developed to model surfactant effects in aqueous droplets of atmospheric relevance, as reviewed by Malila and Prisle (2018). Most approaches either employ Gibbs surface thermodynamics, where the surface-phase is approximated as a two-dimensional interface (e.g., Sorjamaa et al., 2004; Prisle et al., 2008, 2010; Topping, 2010; Raatikainen and Laaksonen, 2011; Petters and Petters, 2016; McGraw and Wang, 2021; Prisle, 2021), or assume a physical surface layer, in the form of a molecular monolayer (e.g., Malila and Prisle, 2018), liquid–liquid phase separation (LLPS) (e.g., Ovadnevaite
et al., 2017), a compressed film, (e.g., Ruehl et al., 2016) or complete phase separation (e.g., Prisle et al., 2011; Ovadnevaite et al., 2017). The different approaches each have specific assumptions and requirements for application. A detailed overview of the history of bulk–surface partitioning is given in the introduction of Vepsäläinen et al. (2022).



Vepsäläinen et al. (2022) investigated six approaches (five bulk–surface partitioning models, developed by Prisle et al. (2010, 2011); Ruehl et al. (2016); Ovadnevaite et al. (2017); Malila and Prisle (2018) and a general bulk solution approach, for reference) to predict droplet activation for moderately surface active organic aerosol. The work showed that the models predict significant differences in the CCN activity, droplet surface tension, and degree of bulk–surface partitioning for particles of malonic, succinic, or, glutaric acid mixed with ammonium sulfate across a range of compositions and the same simulation conditions. However, it is not immediately clear if this result can be generalized to other systems or to different simulation conditions. Therefore, in the current work we use the same six approaches to investigate cloud droplet activation of strongly surface active aerosol under common simulation conditions. The strongly surface active aerosol is represented by either a fatty acid sodium salt (sodium myristate) or the fatty acid itself (myristic acid), mixed with sodium chloride (NaCl) in a wide range of relative mixing ratios. The surface adsorption properties between strong surfactants and moderately surface active compounds are different. We will investigate whether the same conclusions as those found when the models were applied to common systems of moderate surface active compounds (Vepsäläinen et al., 2022) will also hold for strong surfactants.

Strong surfactants have a pronounced ability to reduce surface tension in macroscopic solution and have been found in atmospheric aerosol samples (e.g., Mochida et al., 2002, 2007; Cheng et al., 2004; Li and Yu, 2005; Kourtchev et al., 2013). Fatty acids are a major component of sea spray aerosol (SSA) (e.g., Mochida et al., 2002, 2007; Wang et al., 2015; Cochran et al., 2016; Kirpes et al., 2019) and part of particle compositions associated with ice nucleation by SSA (e.g., DeMott et al., 2018; Perkins et al., 2020). Experiments by Wang et al. (2015) indicate that long-chain fatty acids are the dominant contributor to submicron organic SSA (aerodynamic diameter 0.56 - 1 $\mu$m). Cochran et al. (2016) tentatively identified over 280 organic compounds in nascent SSA, including saturated and unsaturated fatty acids and derivatives of fatty acids. Kirpes et al. (2019) observed thick organic coatings, consisting of marine saccharides, amino acids, fatty acids, and divalent cations, on Alaskan Arctic winter SSA, where 40 % of the particles containing surfactants matched only long-chain fatty acids, while the rest also contained short-chain fatty acids or saccharides.

## 2 Theory and modeling

We consider the cloud droplet activation of surface active aerosol using six different modeling approaches to estimate possible surfactant effects during Köhler calculations of droplet growth. We calculate the Köhler growth curves for particles consisting of sodium myristate (hereafter called NaC14) and NaCl with mass fractions of NaC14 between 0.2 and 0.95 in dry particles of diameter $D_{\mathrm{p}} = 50$ nm. Properties of the individual pure compounds used in the calculations are presented in Table 1. Section S1.1 of the Supplement contains calculations with particles containing myristic acid instead of NaC14 for the models with which we had sufficient preliminary information to perform calculations (simple partitioning, compressed film, and partial organic film models). We describe the strength of surfactants in the same way as in Vepsäläinen et al. (2022), by their ability to reduce the surface tension of aqueous solutions at a given concentration. Descriptions of the different model calculations are briefly given in the following sections. For a more detailed description of the calculations, we refer to Vepsäläinen et al. (2022). Table 1. in Vepsäläinen et al. (2022) gives a summary of the surface tension and water activity calculation methods, as



well as the different compositions of the droplet bulk and surface phases for the different models. Fig. 1 in Vepsäläinen et al. (2022) provides a conceptual figure of the different models. For the most detailed documentation of these models, we refer to the presenting publications of each model.

## 2.1 Köhler theory

Cloud droplet activation in all models is based on equilibrium Köhler theory (Köhler, 1936)

$$S \equiv \frac{p_\mathrm{w}}{p_\mathrm{w}^0} = a_\mathrm{w} \exp\left(\frac{4\bar{v}_\mathrm{w}\sigma}{RTd}\right), \tag{1}$$

where $S$ is the equilibrium water vapor saturation ratio, $p_\mathrm{w}$ is the equilibrium partial pressure of water over the solution droplet, $p_\mathrm{w}^0$ is the saturation vapor pressure over a flat surface of pure water, $a_\mathrm{w}$ is the droplet solution water activity, $\bar{v}_\mathrm{w} = M_\mathrm{w}/\rho_\mathrm{w}$ is the molar volume of water, $\sigma$ is the droplet surface tension, $R$ is the universal gas constant, $T$ is the temperature in Kelvin, and

$d$ is the spherical droplet diameter. Droplet activation is determined in terms of the critical saturation ratio ($S_\mathrm{c}$), or the critical supersaturation ($\mathrm{SS_c}$, $\mathrm{SS} = (S-1)\cdot 100\%$), both corresponding to the maximum value of the Köhler curve described by Eq. (1).

Köhler calculations are initiated by providing the particle dry size and composition. The total amount of NaC14 and NaCl molecules in each particle is calculated with the compound solid-phase densities and relative mass fractions in the particle,

also determining the total amount of solute present in the growing aqueous droplet in the process. All models assume spherical particles and droplets, as well as additive solid-phase volumes of the different compounds. The droplet solution is a ternary mixture of water--inorganic salt--surfactant where the amount of water in the droplets is calculated in one of two ways. The Gibbs, simple partitioning, compressed film, and partial organic film models assume additive volumes between the dry particle and condensed water. In this case, the dry particle volume is subtracted from the total droplet volume (Hänel, 1976).

The monolayer and bulk solution models employ an iterative method based on mass conservation in the droplet, making use of a composition-dependent density function to determine the amount of water in the droplet. More details are provided in Vepsäläinen et al. (2022) and its Supplementary material. The specific treatment of bulk–surface partitioning as well as the corresponding droplet water activity and droplet surface tension vary between the different models as detailed in the following.

## 2.2 Gibbs adsorption partitioning model

In the Gibbs model of Prisle et al. (2010), the Gibbs adsorption equation (Gibbs, 1878) has been combined with the Gibbs–Duhem equation for the droplet bulk, resulting in

$$\sum_j n_j^\mathrm{T} kT \frac{d\ln(a_j^\mathrm{B})}{dn_\mathrm{sft}^\mathrm{B}} + A\frac{d\sigma}{dn_\mathrm{sft}^\mathrm{B}} = 0, \tag{2}$$

where $n_j^\mathrm{T}$ is the total amount of species $j$ in the droplet solution, $k$ is the Boltzmann constant, $n_\mathrm{sft}^\mathrm{B}$ is the number of surfactant ions in the droplet bulk after dissociation (index "sft" generally referring to the surfactant but the solutes are assumed to be

fully dissociated and only the surfactant anions partition to the surface), $a_j^\mathrm{B}$ is the activity of $j$ in the droplet bulk solution, $A$ is



the spherical droplet surface area and $\sigma$ is the droplet surface tension, given as a function of bulk–phase composition. Equation (2) is solved iteratively for the bulk composition with the boundary condition that the molar ratio of water and salt is the same in both the bulk and surface phases, so that the only adsorbing species is the surfactant. We assume volume additivity (such that the droplet diameter is given by the sum of the molar volumes of the individual pure components) and mass conservation

$(n_j^{\mathrm{T}} = n_j^{\mathrm{S}} + n_j^{\mathrm{B}})$ of all components in the droplet (Prisle, 2006). Droplet water activity is calculated as a corrected molar fraction, with both NaCl and NaC14 having a constant dissociation factor of two.

### 2.3 Simple complete partitioning model

The simple partitioning model of Prisle et al. (2011) assumes that surfactants are completely partitioned to the droplet surface into an insoluble layer of pure surfactant. The surface layer is assumed to not affect the kinetics of water condensation–

evaporation equilibrium. The complete partitioning of the surfactant to the surface means that there is no surfactant present in the droplet bulk, which therefore consists of a binary mixture of water and salt. The surface tension of the droplet solution is assumed to be equal to that of pure water, representing that the coverage is insufficient for large droplet surfaces to form a full monolayer (Prisle et al., 2011; Prisle, 2021). The water activity of the water–NaCl solution is calculated through a fit to AIOMFAC (Zuend et al., 2008, 2011; AIOMFAC-web, 2022) calculations as a function of salt mole fraction. The details of

the fit can be found in Sect. S2.1 of the Supplement.

### 2.4 Compressed film surface model

The compressed film model of Ruehl et al. (2016) divides the droplet into an organic surface and a ternary solution droplet bulk of surfactant, inorganic, and water. The droplet water activity is calculated as the corrected molar fraction of the droplet bulk solution. The compressed film equation of state defines the relationship between the surface concentration of the surfactant and

the surface tension in terms of molecular area ($A$)

$$\sigma = \min\left(\sigma_{\mathrm{w}}, \max\left(\sigma_{\mathrm{w}} - (A_0 - A)m_\sigma, \sigma_{\min}\right)\right), \tag{3}$$

where $\sigma_{\mathrm{w}}$ is the surface tension of water, $A_0$ is the critical molecular area, $m_\sigma$ accounts for the interaction between surfactants at the interface, and $\sigma_{\min}$ is a lower limit imposed on the surface tension.

The isotherm for the equation of state (EoS) of the compressed film model is

$$\ln\left(\frac{C_{\mathrm{bulk}}}{C_0}\right) = \frac{(A_0^2 - A^2)m_\sigma N_{\mathrm{A}}}{2RT}, \tag{4}$$

where $C_0$ is the bulk solution concentration at the phase transition, $C_{\mathrm{bulk}}$ is the bulk solution concentration, and $N_{\mathrm{A}}$ is Avogadro's number. The bulk concentration of surfactant and molecular area can be expressed as functions of different diameters (salt seed ($D_{\mathrm{seed}}$), dry particle ($D_{\mathrm{p}}$) and droplet ($d$)) and the fraction of organic molecules partitioned to the surface ($f_{\mathrm{surf}}$) as

$$C_{\mathrm{bulk}} = \frac{(1 - f_{\mathrm{surf}})(D_{\mathrm{p}}^3 - D_{\mathrm{seed}}^3)\bar{v}_{\mathrm{w}}}{d^3 \bar{v}_{\mathrm{sft}}} \tag{5}$$



and

$$A = \frac{6\bar{v}_{\text{sft}}d^2}{f_{\text{surf}}(D_{\text{p}}^3 - D_{\text{seed}}^3)N_{\text{A}}}. \tag{6}$$

The values used for the model specific parameters $A_0, C_0, m_\sigma$ and $\sigma_{\min}$ were obtained from Forestieri et al. (2018) for myristic acid. We use the same values for NaC14 due to lack of suitable data to fit the parameters for NaC14. The molar volumes of the acid and its sodium salt are different, as is the dissociation factor. Therefore, an analysis of the differences between the

predictions with NaC14 and myristic acid as well as a sensitivity analysis of the compressed film model with respect to the molar volume of the surfactant have been provided in Sects. S1.1 and S1.2 of the Supplement. Predictions with myristic acid present in the particles lead to higher predicted critical supersaturations and smaller predicted critical droplet diameters than with particles containing NaC14. Varying the molar volume of myristic acid affects the predictions of the compressed film model more at high surfactant mass fractions. Decreasing the molar volume by roughly 25 %, leads to predictions of

the critical point similar to those of NaC14 at surfactant mass fraction of $w_{\text{p,sft}} = 0.95$ in the particles. The Supplement of Vepsäläinen et al. (2022) contains a more thorough sensitivity analysis of all models with respect to various parameters. The model parameters used here are $A_0 = 29.2\text{ Å}^2, \log_{10}C_0 = -7.4, m_\sigma = 1.28\text{ mJ m}^{-2}\text{ Å}^{-2}$. The minimum surface tension model parameter $\sigma_{\min}$ was assigned a value of zero as a conservative estimate because it could not be determined experimentally (Forestieri et al., 2018). These model specific parameters $A_0, C_0, m_\sigma$ and $\sigma_{\min}$ are assumed to be compound specific physical

constants and therefore not sensitive to seed or coated particle diameters or the droplet dilution state, such that they can be applied across a range of particle sizes and compositions.

## 2.5   Partial organic film model

In the partial organic film model of Ovadnevaite et al. (2017), all organic content resides in a salt and water-free surface film that is assumed to completely coat the droplet bulk until a minimum surface thickness is reached ($\delta_{\text{sft}}$) and the surface film

breaks, resulting in partial coverage of the droplet. The water activity of the droplet bulk consisting only of salt and water is calculated through a fit to AIOMFAC (Zuend et al., 2008, 2011; AIOMFAC-web, 2022) calculations as a function of salt mole fraction in a range relevant to the calculations. See Sect. S2.1 of the Supplement for details.

The surface tension of an individual liquid phase is calculated as a volume fraction–weighted mean of the pure-component surface tension values ($\sigma_j$). For the droplet bulk

$$\sigma^{\text{B}} = \sum_j \varphi_j^{\text{B}}\sigma_j, \tag{7}$$

where $\varphi_j^{\text{B}}$ is the volume fraction of the component $j$ in the bulk-phase ($\sum_j \varphi_j^{\text{B}} = 1$). The parameter that describes the surface coverage $c_{\text{S}}$ is defined as

$$c_{\text{S}} = \min\left(\frac{V^{\text{S}}}{V^\delta}, 1\right). \tag{8}$$

The organic film covers the droplet bulk completely ($c_{\text{S}} = 1$) or partially ($c_{\text{S}} < 1$). $V^{\text{S}}$ is the volume of the surface-phase at

droplet diameter $d$ and $V^\delta$ is the corresponding volume of a spherical shell of thickness $\delta_{\text{sft}}$, which is the minimum surface





thickness before the organic surface film breaks. The surface thickness $\delta_{\text{sft}}$ is set equal to the surface thickness values calculated by the monolayer model of Malila and Prisle (2018) for the same test system. Originally, Ovadnevaite et al. (2017) assumed a single constant value of the surface thickness, but since the monolayer model already explicitly calculates surface thickness, here we use these values to improve the consistency between the results of the different models. Sect. S1.3 of the Supplement

contains a comparison between using a constant value as $\delta_{\text{sft}}$, and using the values calculated with the monolayer model.

The effective surface tension of the droplet is calculated as the surface–area–weighted mean of the surface tensions from both phases as

$$\sigma = (1 - c_{\text{S}})\sigma^{\text{B}} + c_{\text{S}}\sigma^{\text{S}}. \tag{9}$$

### 2.6 Monolayer model

In molecular monolayer model of Malila and Prisle (2018) the partitioning between the bulk and surface phases for each compound $j$ is calculated iteratively from an extension of the Laaksonen–Kulmala equation (Laaksonen and Kulmala, 1991)

$$\sigma(x^{\text{B}}, T) = \frac{\sum_j \sigma_j v_j x_j^{\text{S}}}{\sum_j v_j x_j^{\text{S}}}, \tag{10}$$

where $v_j$ is the liquid phase molecular volume, $\sigma_j$ the surface tension, while $x_j^{\text{S}}$ and $x_j^{\text{B}}$ are the droplet surface and bulk mole fractions of compound $j$ respectively. The condition of mass conservation ($n_j^{\text{T}} = n_j^{\text{S}} + n_j^{\text{B}}$) is imposed on the calculation of

each compound $j$. The droplet water activity is calculated as the corrected molar fraction using the composition of the droplet bulk after the bulk–surface partitioning. The thickness of the surface monolayer is calculated as

$$\delta = \left( \frac{6}{\pi} \sum_j v_j x_j^{\text{S}} \right)^{1/3}. \tag{11}$$

### 2.7 The bulk solution model

The results of the different bulk–surface partitioning models are compared with a bulk solution model where the droplet

properties are assumed to be equivalent to those for a macroscopic solution with the same total composition. In particular, bulk–surface partitioning does not affect the concentration of surfactants in the droplet bulk solution. The droplet bulk is equivalent in volume to the whole droplet, and no separate surface-phase exists. The droplet surface tension is estimated as a function of composition through a fit made into the ternary data of Wen et al. (2000) for mixtures of NaCl, NaC14, and water. The surface tension above the CMC is a constant value. The details of the fit are presented in Sect. S2.2 of the Supplement.

Water activity is calculated as a corrected molar fraction, assuming that the critical micelle concentration (CMC) limits the maximum amount of surfactant dissolved in the bulk. Any additional surfactant is assumed to be undissolved and has no impact on the calculation.





**Table 1.** The molar masses ($M$), the densities of the liquid and solid phases ($\rho_l$ and $\rho_S$) and the surface tensions ($\sigma$) of the different pure compounds.

| Compound | $M$ (g mol$^{-1}$) | $\rho_l$ (kg m$^{-3}$) | $\rho_s$ (kg m$^{-3}$) | $\sigma$ (mN m$^{-1}$) |
|---|---|---|---|---|
| Water | 18.0153 | 997.05[a] | - | 71.97[b] |
| NaCl | 58.4428 | 1977.1238[c] | 2165[d] | 169.7398 [c] |
| Sodium myristate (NaC14) | 250.353 | 1039.7[e] | 1200[f] | 24.2[g] |

[a] Pátek et al. (2009), [b] International Association for the Properties of Water and Steam (IAPWS) (2014), [c] Vanhanen et al. (2008); Janz (1980), [d] National Toxicology Program (1993), [e] Extended from binary aqueous density estimated via method of Calderón and Prisle (2021), [f] Estimate (Prisle et al., 2008) [g] Value at CMC

## 2.8 Critical micelle concentration

Above the so-called critical micelle concentration (CMC), some surfactants can self-aggregate to form structures here collec-
tively referred to as *micelles*. The micellization process is highly dependent on the surfactant species. (Langevin, 1992)

In this work, the critical micelle concentration in the droplet bulk is assumed equal to that of the binary solution of water–
NaC14 determined graphically from the digitized surface tension measurements of Wen et al. (2000). More information is
available in Sect. S2.4 of the Supplement. The surface tension after the CMC is reached is assumed to be constant, and is
determined by the binary organic surface tension fit to be $\sigma_{\mathrm{CMC}} = 24.2$ mN m$^{-1}$. We are aware that in ternary solutions
of water-surfactant-inorganic salt, both the CMC and the surface tension at the CMC vary depending on the inorganic salt
concentration. These variations also depend on the chemical nature of the inorganic salt (i.e., whatever inorganic salt and the
ionic surfactant share the surfactant counterion as a common ion). However, accounting for these effects requires the use of
complex modeling frameworks (e.g., Kralchevsky et al., 1999) with parameters that must be retrieved from experiments that
are scarce or nonexistent for these ternary systems.

Due to lack of available data on the actual pure compound surface tension (in a hypothetical supercooled liquid state), the
binary surface tension at the CMC is assumed to be equal to the surface tension of pure NaC14 when such a value is needed
(in the monolayer and partial organic film models). The CMC is taken into account in the composition-dependent ternary
surface tension function used for the monolayer, Gibbs, and bulk solution models, ensuring that the droplet surface tension
does not assume values smaller than the value at the CMC ($\sigma_{\mathrm{CMC}}$). After the main calculations of the monolayer model are
performed, a check is made such that if the predicted surface tension is equal to the value at the CMC, the droplet bulk and
surface compositions are assumed to be unknown. The comparison is made with the assumption that the salt has no effect on
the surface tension value at the CMC. This method is different from what was originally described in Malila and Prisle (2018)
where the surfactant surface mole fraction equal to unity was assumed, when the bulk surfactant concentration was above the
CMC value. We employed a similar check for the Gibbs model, setting the droplet composition to unknown if the predicted
droplet surface tension value is that at the CMC. The water activity calculated with the bulk solution model also employs this





check with the predicted droplet surface tension to determine if surfactant concentration in the droplet is above the CMC. The maximum amount of surfactant dissolved in the bulk is limited by the CMC.

## 3 Results and discussion

In the following sections, we present the results of the simulations performed with the different models detailed above for
common particle systems containing NaC14 and NaCl. Section S1.1 of the Supplement presents analogous results for particles containing myristic acid.

### 3.1 Köhler curves and droplet activation

Figure 1 shows the Köhler curves predicted with the different models in terms of supersaturation (SS) for dry particles with $D_\mathrm{p} = 50$ nm and at NaC14 mass fractions ($w_\mathrm{p,sft}$) of 0.2, 0.5, 0.8 and 0.95. The critical points ($d_\mathrm{c}$, $\mathrm{SS_c}$) are shown in Table
2 and are determined as the maximum SS value of each curve with the corresponding droplet diameter. Fig. 1 immediately highlights that the different models can predict significantly different critical points for the same dry particle systems. Of the partitioning models, the simple partitioning and compressed film models consistently predict the highest $\mathrm{SS_c}$ values and the smallest $d_\mathrm{c}$ values. The Gibbs model predicts larger $d_\mathrm{c}$ values than the other partitioning models and $\mathrm{SS_c}$ values between the extremes of the predictions. The monolayer model and the partial organic film model predictions agree well across all NaC14
mass fractions, and the predictions of the critical point with the two models are always between the maximum and minimum predictions of the six models. As seen in numerous previous studies (e.g., Sorjamaa et al., 2004; Prisle et al., 2008, 2010; Topping, 2010; Prisle et al., 2019; Prisle, 2021), there is also a large difference between the predictions of the partitioning models and the bulk solution model, especially in terms of supersaturation. The predictions of the critical droplet properties with the different bulk–surface partitioning models begin to show more significant differences in both $\mathrm{SS_c}$ and $d_\mathrm{c}$ as the NaC14
mass fraction in the particles increases. Furthermore, in Fig. 1(d) for particles containing NaC14 ($w_\mathrm{p,sft} = 0.95$), none of the model predictions match the $\mathrm{SS_c}$ value fitted to experimental data (Prisle et al., 2008).

In Fig. 1(a) with $w_\mathrm{p,sft} = 0.2$, the different curves and $\mathrm{SS_c}$ values predicted by the different models are comparable, apart from the predictions of the bulk solution model. The monolayer and partial organic film models predict very similar $\mathrm{SS_c}$ even as the NaC14 mass fraction increases (Fig. 1 and Table 2). We used the surface thickness ($\delta$) predicted by the monolayer model as
input for the partial organic film model calculations, but this contributes only a small enhancement to the similarities between the predictions of the two models. Using a constant surface thickness of 0.5 nm with the partial organic film model does not drastically change the SSpredictions $_\mathrm{c}$ (a maximum of 4 % calculated from $\mathrm{SS_c^{\delta_{ML}}}/SS_\mathrm{c}^{\delta_{0.5}}$ for $w_\mathrm{p,sft} = 0.5$ and 0.8). See Sect. S1.3 of the Supplement for the full comparison.

The Gibbs model predicts $\mathrm{SS_c}$ values comparable to the monolayer and partial organic film models for all particle composi-
tions in Fig. 1, but the critical point predicted with the Gibbs model is always immediately after the dilution state of the growing droplet is sufficient to overcome the CMC. This could be due to a numerical artifact of the Gibbs model caused by the CMC constraint imposed on the simulation. As none of the models employed accounts for the CMC variation with NaCl content in



the droplet solution, nor for the explicit effect of micelles in the thermodynamic expression for the adsorption equilibrium, any explanation given to the Gibbs model response is speculative.

The simple partitioning and compressed film predictions of $SS_c$ increase considerably more with the NaC14 mass fraction than with the other models (Table 2). In Figs. 1(a )- 1(c), the simple partitioning model always predicts slightly higher $SS_c$. In Fig. 1(d), the difference between the predicted $SS_c$ values of the two models increases. As expected with particles containing a strong surfactant, the bulk solution model predicts lower $SS_c$ values than any of the partitioning models, as has been extensively observed in previous work (e.g., Sorjamaa et al., 2004; Prisle et al., 2008, 2010; Topping, 2010; Prisle et al., 2019; Prisle, 2021).

The bulk solution model did not predict significantly lower $SS_c$ values compared to the bulk–surface partitioning models for particles containing malonic, succinic, or glutaric acid and ammonium sulfate due to the moderately surface active nature of the three acids (Vepsäläinen et al., 2022).

The $SS_c$ predictions for particles containing NaC14 ranges from 0.08 % to 1.63 % (see Table 2). Average supersaturations in low-level clouds range from 0.1 % to 0.4 % (e.g. Politovich and Cooper, 1988). Higher values of 0.7 % to 1.3 % can be reached

during strong convection (e.g. Siebert and Shaw, 2017; Yang et al., 2019). Only the simple partitioning and compressed film models predict values above 0.7 % at high surfactant mass fractions, while the other models predict $SS_c$ values feasible for low-level clouds. The absolute differences between the highest and lowest calculated values for $SS_c$ in Fig. 1 are $\Delta SS_c = 0.27, 0.42, 0.73$, and $1.49$ % for $w_{p,sft} = 0.2, 0.5, 0.8$ and $0.95$, respectively. However, the bulk solution model is an outlier in most cases. Among the bulk–surface partitioning models, then the maximum differences between the predicted critical

supersaturation values are $\Delta SS_c = 0.10, 0.24, 0.55$, and $1.29$ % for $w_{p,sft} = 0.2, 0.5, 0.8$ and $0.95$, respectively.

An experimentally derived $SS_c$ value from Prisle et al. (2008) for pure NaC14 particles ($SS_c^{exp} = 0.9655\%$) is included in Fig. 1(d). The $SS_c^{exp}$ value for pure NaC14 particles of $D_p = 50$ nm falls between the extremes of the different model predictions. The simple partitioning and compressed film model predictions of $SS_c$ are larger, while the monolayer, Gibbs, partial organic film, and bulk solution model predictions are smaller than $SS_c^{exp}$. None of the different model predictions matches the

experimental value, even when considering that the comparison is between $w_{p,sft} = 1$ and $w_{p,sft} = 0.95$. Simulations with pure NaC14 particles have a somewhat higher $SS_c$ than with $w_{p,sft} = 0.95$, but the difference is minor for the predictions of the monolayer, Gibbs, and partial organic film models compared to the large discrepancy between the model predictions and $SS_c^{exp}$ shown in Fig. 1(d). Predictions of the simple model have been observed to overestimate $SS_c$ at high surfactant mass fractions in the past (Prisle et al., 2011; Vepsäläinen et al., 2022). The droplet surface tension parametrization used for this work is based

on a different data set for an aqueous system containing NaC14 and NaCl (Wen et al., 2000) than the one employed by Prisle et al. (2008) for aqueous solutions of NaC14 during their simulations (Campbell and Lakshminarayanan, 1965). The disparity in the predicted droplet surface tension based on the two different fits explains the difference in the predictions between the Gibbs model employed in Prisle et al. (2008) and the predictions with the Gibbs model in the present work.

The critical diameter ($d_c$) values predicted with the different models in Fig. 1 decrease mainly as the NaC14 mass fraction in

the particles increases, excluding the predictions of the Gibbs model in Figs. 1(a) and (b) where the situation is reversed and $d_c$ increases with $w_{p,sft}$. The Gibbs model predicts the largest $d_c$ values out of all the bulk–surface partitioning models in Fig. 1, and generally even when including the predictions of the bulk solution model outside of Fig. 1(a). However as the critical point





for cloud droplet activation predicted with the Gibbs model is directly related to the droplet size where dilution overcomes the CMC, these values predicted with a constant CMC limit may not be reliable.

The predictions of the monolayer, compressed film, and partial film models of $d_c$ are closely grouped in Fig. 1(a) but the monolayer and partial film model predict similar $d_c$ for all particle compositions (Fig. 1, Table 2) . The difference in $d_c$ between using the surface thickness predicted by the monolayer model and a constant value of 0.5 nm with the partial film model is a maximum of 8 % for $w_{p,sft} = 0.95$. The full comparison is presented in Sect. S1.3 of the Supplement. The simple partitioning and compressed film models predict similar $d_c$ values that decrease as the NaC14 mass fraction in the particles increases, only

Fig. 1(d) showing a significant difference. The curve predicted by the bulk solution model has a distinct shape in the form of two local maxima in Fig. 1, because the predicted droplet surface tension is at a constant CMC value before eventually starting to increase due to the dilution of the droplet solution as the droplet grows (Fig. 2). The bulk solution model in Fig. 1(a) predicts considerably larger $d_c$ than the other models, corresponding to predictions where the critical point is reached at a surface tension other than the value at the CMC. The absolute differences between the largest and smallest calculated values

for $d_c$ in Fig. 1 are $\Delta d_c = 947.84, 479.56, 509.86$, and $518.29$ nm % for $w_{p,sft} = 0.2, 0.5, 0.8$ and $0.95$, respectively. Excluding the $d_c$ predictions of the bulk solution model only affects the maximum difference in Fig. 1(a), where $\Delta d_c = 337.09$ nm for $w_{p,sft} = 0.2$ between only the bulk–surface partitioning models.

The inter-model comparison of predicted Köhler curves in Fig. 1 for particles containing NaC14 does not follow the same pattern as previously observed for malonic, succinic, or glutaric acid (Vepsäläinen et al., 2022). In Fig. 1, the simple partitioning

and the compressed film models predict the highest $SS_c$ at high mass fractions of NaC14, the monolayer and partial film models agree well for the entire range of particle compositions, while the Gibbs model predicts comparable $SS_c$. For malonic, succinic, and glutaric acids, the simple partitioning and partial film models predicted the highest $SS_c$, the Gibbs, monolayer, and bulk solution models predicted similar droplet properties, while the compressed film model predicted comparable $SS_c$. The Gibbs model predictions depend strongly on the CMC for droplets containing NaC14, which is a significant change from malonic,

succinic, and glutaric acids, as the three acids are not known to form micelles. Predictions of the partial organic film model at $w_{p,sft} = 0.8$ and $0.95$ for NaC14 also do not show the same reduction in $d_c$ as with malonic, succinic, and glutaric acids. This is a result of the critical point in Figs. 1(c) and 1(d) being reached only after the organic surface film has broken and covers the droplet bulk only partially. The droplet size where the organic film breaks is not clearly visible in Fig. 1, but we assume there is no change in the relative location of the critical point even if the smallest droplet sizes could be plotted based on Fig. S1 of

the Supplement, where the simulation with a constant surface thickness of 0.5 nm for particles containing NaC14 is displayed, and no change in the relative location of the critical point is observed.

## 3.2 Droplet surface tension

Figure 2 shows the droplet surface tensions along the Köhler curves calculated with the different models for initial dry particles of $D_p = 50$ nm and at NaC14 mass fractions ($w_{p,sft}$) of 0.2, 0.5, 0.8 and 0.95. Table 2 contains the droplet surface tensions at

the critical point of cloud droplet activation predicted with the different models for each particle system. In Fig. 2, all panels show similar characteristics in the overall structures of the predicted droplet surface tension curves. The simple partitioning





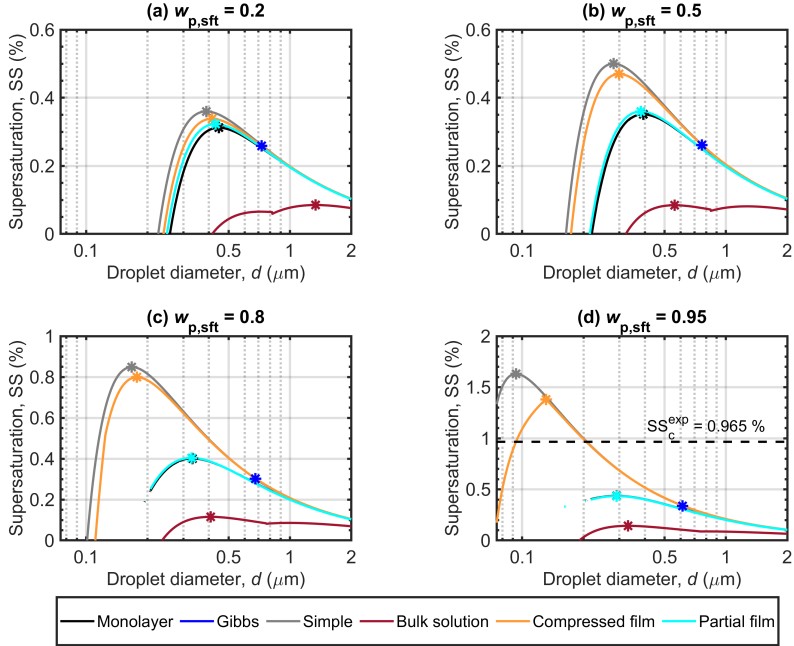

**Figure 1.** Köhler curves calculated with the different models for dry NaC14–NaCl particles with $D_\mathrm{p} = 50$ nm. Each panel shows curves for particles with different NaC14 mass fractions ($w_\mathrm{p,sft}$). Critical points are also marked on each curve, and the experimental critical supersaturation reported by Prisle et al. (2008) is included in panel (d) corresponding to $w_\mathrm{p,sft} = 1$. Note that the vertical-axis scaling changes between the panels.

model assumes a constant surface tension value equal to that of water. Predictions of the droplet surface tension during droplet growth vary significantly between the different models. Surface tension depression is significant at small droplet sizes for most predictions of the bulk–surface partitioning models, except for the compressed film model in Fig. 2(a). At the point of droplet

activation, the predicted degree of surface tension depression varies significantly between the different models and different particle compositions. Differences in the predicted surface tension at activation increase together with the NaC14 mass fraction in the particles. In Fig. 2(d) the predicted surface tension at droplet activation varies from the CMC value to the surface tension of water between the different models. In Fig. 2, the monolayer and partial organic film models predict a degree of surface tension depression that increases with the NaC14 mass fraction in the particles, while the Gibbs model only predicts a small

depression from the surface tension of water. The compressed film model predicts the surface tension of water at the point of cloud droplet activation for all particle compositions. The predictions of the bulk solution model are equal to the surface tension at the CMC in all panels of Fig. 2 except (a), but the predicted surface tension depression is still the largest of the models.

      The surface tension predicted with the Gibbs model has a step increase from the surface tension at CMC to a surface tension near that of water for all panels of Fig 2. The droplet diameter at the step increase (or immediately after it) corresponds to

the critical point for all panels of Fig. 2, as mentioned above during the discussion on the predicted Köhler curves. The step




increases in Fig. 2 are visible for the benefit of the reader, as the droplet composition is assigned to be unknown when the droplet surface tension is equal to the surface tension at CMC. The surface tension curves predicted with the monolayer and partial organic film models are similar for all panels of Fig. 2. The partial organic film model calculations were also tested with a constant $\delta = 0.5$ nm, but the predictions with a constant surface thickness show only minor changes compared to the

curves presented here. Detailed results of the differences are presented in Sect. S1.3 of the Supplement. The breaking of the surface organic film in the predictions of the partial organic film model is not visible in the figures presented, aside from Fig. 2(a) where the film breaks near the smallest visible droplet sizes.

The surface tension predictions of the compressed film model in Fig. 2 generally display behavior different from previous studies featuring the model (e.g., Ruehl et al., 2016; Forestieri et al., 2018; Vepsäläinen et al., 2022). The compressed film

model predicts no visible surface tension depression in Fig. 2(a). Surface tension depression before activation is visible in Figs. 2(b) - (d), but the critical point is at the same droplet size as the point when the surface tension reaches that of water only in Fig. 2(d) as was the typical behavior in Ruehl et al. (2016). Vepsäläinen et al. (2022) observed an occurrence of the critical point and surface tension becoming that of water at different points with carboxylic acids and ammonium sulfate particles at $w_{\mathrm{p,org}} = 0.2$ for $D_{\mathrm{p}} = 50$ nm. Forestieri et al. (2018) observed the same for 80 nm NaCl particles coated with oleic acid at organic volume

fraction of 0.8. We interpret the difference in observed behavior of the surface tension predictions between Fig. 2 and past works to indicate that the model specific parameters used in this work do not adequately capture the solution properties of the aqueous droplet solution containing NaC14 and NaCl. This is not surprising as the parameters used in this work were fitted for particles containing myristic acid by Forestieri et al. (2018), not NaC14. Simulations comparing the predictions between NaC14 and myristic acid in Sect. S1.1 of the Supplement reveal that the critical point and the surface tension reaching that

of water do not happen at the same droplet size even for myristic acid. Forestieri et al. (2018) fitted the model parameters at organic volume fraction range of $0.40 - 0.98$ for $D_{\mathrm{seed}} = 180, 200$ nm and RH range of 99.83-99.93 %. This suggests that the model is sensitive to the specific fitting conditions of the model parameters, and the assumption that the model parameters are compound specific physical constants across varying NaC14 mass fractions and dry particle sizes may not hold true for real non-ideal droplet solutions. The fitting of the model parameters of the compressed film model assumes that the droplet solution

is pseudo-ideal, but systems of fatty acids or sodium salts of fatty acids can show significant deviation from an ideal solution (Michailoudi et al., 2020; Calderón et al., 2020). In Fig. 2, the droplet surface tension predicted with the compressed film model is always above the surface tension at the CMC of the binary mixture of NaC14–water that has been used to denote the physical lower limit of the droplet surface tension. The lowest predictions of droplet surface tension during droplet growth for malonic, succinic, or glutaric acids with the compressed film model were lower than the surface tension of the pure compound

due to the model parameter $\sigma_{\mathrm{min}}$ (Vepsäläinen et al., 2022).

In Fig. 2, generally the predicted surface tension values at the critical point of cloud droplet activation ($\sigma_{\mathrm{c}}$) decrease as the NaC14 mass fraction in the particles ($w_{\mathrm{p,sft}}$) increases. At $w_{\mathrm{p,sft}} = 0.2$ in Fig. 2(a), the minimum $\sigma_{\mathrm{c}}$ value predicted between the monolayer, Gibbs, compressed film and partial organic film models is 67.27 mNm$^{-1}$ (Table 2), within 4.7 mNm$^{-1}$ of surface tension of water. In Fig. 2(b), the maximum difference increases to 11.95 mNm$^{-1}$, in Fig. 2(c) 20.47 mNm$^{-1}$ and in

Fig. 2(d) 26.86 mNm$^{-1}$. This large depression is only observable with the predictions of the monolayer and partial organic



film model. Both models have also predicted significant surface tension depression in previous work (Ovadnevaite et al., 2017; Malila and Prisle, 2018; Lin et al., 2018, 2020; Vepsäläinen et al., 2022). The simple partitioning and compressed film models predict $\sigma_c$ equal to the surface tension of water for all particle compositions. The Gibbs model predicts $\sigma_c$ to be very close to that of water in all cases, but predicts a small degree of surface tension depression (Table 2). The Gibbs and compressed film

model predictions of a small degree of surface tension depression or no surface tension depression are in agreement with past works (Prisle et al., 2008, 2010; Ruehl et al., 2016; Forestieri et al., 2018; Lin et al., 2018; Prisle, 2021; Vepsäläinen et al., 2022). The Gibbs model has been observed to predict higher surface tension values than the monolayer model to a significant degree for SDS-NaCl and ragweed-ammonium sulfate particles, as well as somewhat lower surface tension than the monolayer model for succinic acid-NaCl particles (Lin et al., 2018). The bulk solution model predicts very low $\sigma_c$ values for all particle

compositions. In Fig. 2(a), the predictions of the bulk solution model reach the critical point after the dilution is sufficiently strong to move away from the surface tension value at the CMC, but the reduction of the surface tension of water is still 27.96 mNm$^{-1}$. In Figs. 2(b) - 2(d), the critical points are reached when the surface tension is at the value of the CMC surface tension for the bulk solution model predictions, a 47.72 mNm$^{-1}$ reduction from the surface tension of water.

The predicted droplet surface tension in Fig. 2 for particles containing NaC14 are significantly different from the curves

for droplets containing malonic, succinic, or glutaric acid in Vepsäläinen et al. (2022). In Fig. 2 the monolayer, Gibbs, and bulk solution models predict different surface tensions, although they use the same surface tension function. The three models predicted similar surface tensions for malonic, succinic, and glutaric acids (Vepsäläinen et al., 2022). The step increase in surface tension observed in Fig. 2 with the Gibbs model predictions is a direct consequence of the CMC, as is the extremely low surface tension predicted with the bulk solution model for a significant portion of the droplet growth. This is in clear

contrast to Vepsäläinen et al. (2022), as malonic, succinic, and glutaric acids are not known to form micelles. Generally, the predictions for $\sigma_c$ for particles containing NaC14 are much lower than those for particles containing malonic, succinic, or glutaric acid with the models where surface tension depression is possible and with the exception of the partial organic film model, which predicts similar droplet surface tension values for both strong surfactant and moderately surface active organics.

### 3.3   Surfactant bulk–surface partitioning

Figure 3 shows the surface partitioning factors of NaC14 in the droplets, calculated as the fraction of NaC14 molecules present in the droplet surface compared to the total amount of NaC14 in the droplet ($n_{sft}^S/n_{sft}^T$), with the different models for NaC14-NaCl dry particles of $D_p = 50$ nm and at NaC14 mass fractions ($w_{p,sft}$) of 0.2, 0.5, 0.8 and 0.95. The simple partitioning model and the partial organic film model calculations are performed with the assumption that all organic content is partitioned to the droplet surface during the entirety of the droplet growth and the partitioning factor is equal to unity. The bulk solution

model has no bulk–surface partitioning, and the partitioning factor is zero. Because of this, these constant values of the simple partitioning, partial film, and bulk solution model partitioning factors are not shown in Fig. 3.

Figure 3 shows that the different partitioning models predict very different compositions of surfaces for growing droplets formed on the same aerosol systems. The predicted bulk–surface partitioning of NaC14 is strongest with the compressed film model, followed by the Gibbs and monolayer models, respectively. The predicted partitioning factor value is near or



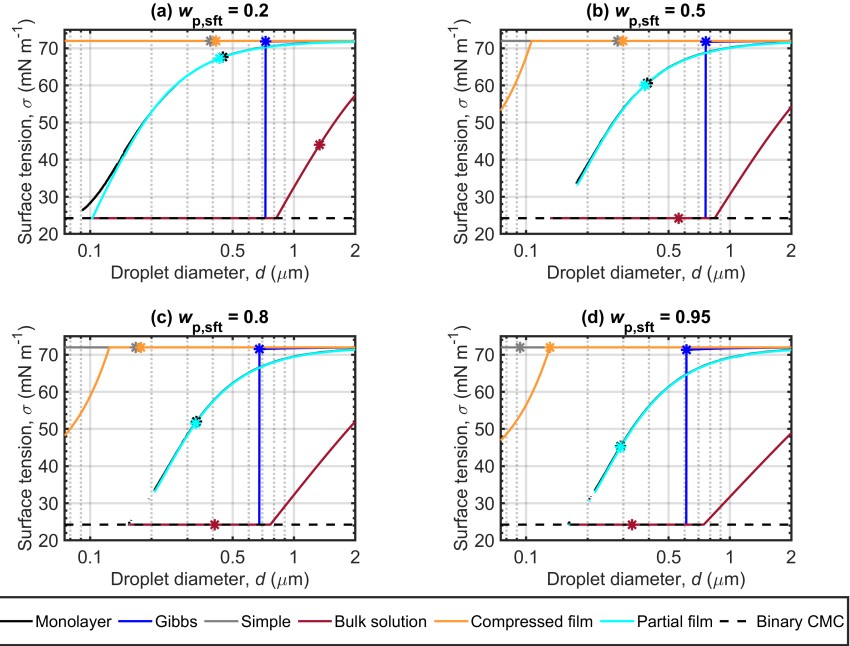

**Figure 2.** Surface tensions of droplets along the Köhler curves, calculated with the different models for dry particles of $D_p = 50$ nm at different NaC14 mass fractions ($w_{p,sft}$). The critical points evaluated for the Köhler curves in Fig. 1 are also marked. The surface tension for the binary mixture of water - NaC14 has been estimated from the measurements of Wen et al. (2000), and is indicated as a physical lower limit for the droplet surface tension.

equal to unity with the three models throughout the droplet growth, meaning majority or all NaC14 content is residing in the droplet surface. The partitioning factors predicted with the monolayer and Gibbs models decrease slightly as the droplet grows. Fig. 3 shows the partitioning factors predicted with the Gibbs model only after the droplet has grown sufficiently to have a composition where micelles no longer form. For NaC14, the degree of NaC14 partitioning is similar between the Gibbs, compressed film, and monolayer models predictions, but the predicted activation properties can vary significantly (Table 2).

For malonic, succinic, and glutaric acids, the compressed film model predicted considerably larger partitioning factors, and there was some variation between the Gibbs and monolayer model predictions but the different models could still predict similar droplet activation properties (Vepsäläinen et al., 2022). The monolayer and Gibbs models predict significantly higher partitioning factors for NaC14 compared to succinic acid or SDS, both mixed with NaCl, in Lin et al. (2018). In the same work, the Gibbs model did predict high partitioning for Nordic aquatic fulvic acid (NAFA) and ragweed mixed with NaCl

and ammonium sulfate, respectively. The monolayer model did also predict high partitioning factor for ragweed but the factor decreased significantly as the surfactant mass fraction in the particles approached unity. The Gibbs model also predicted generally higher partitioning factors compared to the monolayer model in Lin et al. (2018). Lin et al. (2020) used a Gibbs model and a monolayer to predict bulk–surface partitioning of NAFA-NaCl particles, but the partitioning factor with the Gibbs





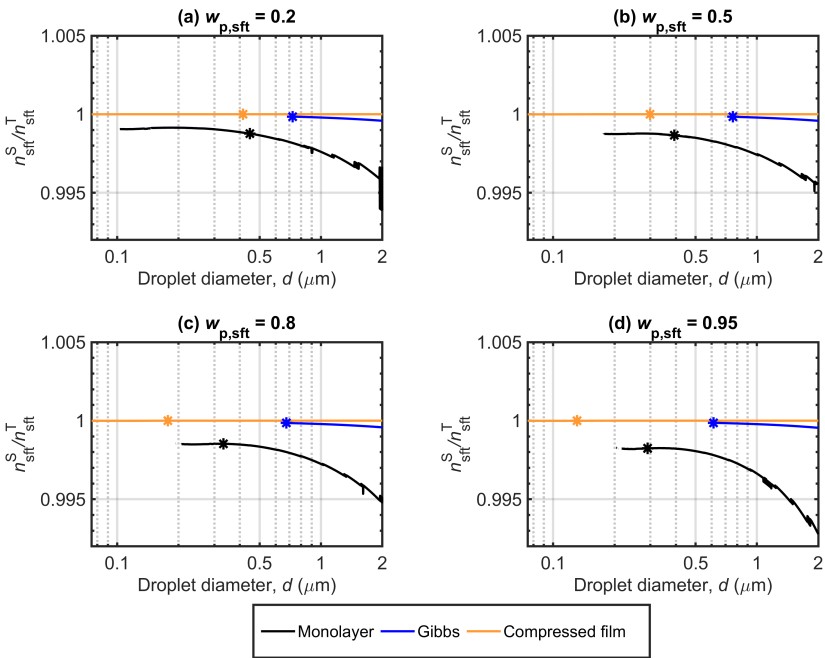

**Figure 3.** NaC14 partitioning factors ($n_{\text{sft}}^{\text{S}}/n_{\text{sft}}^{\text{T}}$) predicted with the monolayer, Gibbs, and compressed film models along the Köhler curves for dry particles with $D_{\text{p}} = 50$ nm at different NaC14 mass fractions ($w_{\text{p,sft}}$). The critical points are also marked.

model was defined differently from Fig. 3 and direct comparison is not possible. Lin et al. (2020) used the monolayer model to

predict moderate to high partitioning factors (roughly 0.3-0.9) in a range of particle sizes ($D_{\text{p}} = 50-150$ nm) and compositions ($w_{\text{p,org}} = 0.02-1$) while accounting for surface tension time evolution. Comparing the predictions of the compressed film model partitioning factor for NaC14 with those of Ruehl et al. (2016) for carboxylic acids, the minimum partitioning factors during droplet growth predicted for NaC14 are equal to unity, while those predicted in Ruehl et al. (2016) generally only increase to unity at the critical point of cloud droplet activation.

**4 Conclusions**

We have used five bulk–surface partitioning models (Prisle et al., 2010, 2011; Ruehl et al., 2016; Ovadnevaite et al., 2017; Malila and Prisle, 2018) and a general bulk solution model in predictive Köhler modeling of growing aqueous solution droplets for identical particles consisting of a strong surfactant mixed with NaCl across a range of particle compositions.

The different models can predict significantly different properties of the activating droplets ($\text{SS}_{\text{c}}$, $d_{\text{c}}$ and $\sigma_{\text{c}}$) for identical

particles. The differences between the bulk–surface partitioning models increase with the surfactant mass fraction in the particles. All bulk–surface partitioning models predict or assume a high degree of bulk–surface partitioning of NaC14, leading to the Raoult effect having only a small effect between the different predictions. The differences in the predicted critical droplet





**Table 2.** The critical droplet diameters ($d_\mathrm{c}$), supersaturations ($SS_\mathrm{c}$) and surface tensions ($\sigma_\mathrm{c}$) for the different models for simulations with NaC14 at 298.15 K for $D_\mathrm{p}$ = 50 nm.

| Parameter | $d_\mathrm{c}$ (nm) | $SS_c$ (%) | $\sigma_\mathrm{c}$ (mN m$^{-1}$) | $d_\mathrm{c}$ (nm) | $SS_c$ (%) | $\sigma_\mathrm{c}$ (mN m$^{-1}$) | $d_\mathrm{c}$ (nm) | $SS_c$ (%) | $\sigma_\mathrm{c}$ (mN m$^{-1}$) |
|---|---|---|---|---|---|---|---|---|---|
| $w_\mathrm{p,sft}$ | | Monolayer | | | Gibbs | | | Simple | |
| 0.2 | 447.41 | 0.31 | 67.62 | 726.17 | 0.26 | 71.84 | 389.09 | 0.36 | 71.97 |
| 0.5 | 393.52 | 0.35 | 60.6 | 760.65 | 0.26 | 71.7 | 281.09 | 0.5 | 71.97 |
| 0.8 | 332.27 | 0.4 | 51.98 | 677.33 | 0.3 | 71.49 | 167.47 | 0.85 | 71.97 |
| 0.95 | 291.05 | 0.44 | 45.37 | 611.67 | 0.34 | 71.31 | 93.38 | 1.63 | 71.97 |
| $w_\mathrm{p,sft}$ | | Bulk solution | | | Compressed film | | | Partial organic film | |
| 0.2 | 1336.92 | 0.08 | 44.01 | 414.51 | 0.34 | 71.97 | 432.19 | 0.32 | 67.27 |
| 0.5 | 559.18 | 0.08 | 24.25 | 298.6 | 0.47 | 71.97 | 382.52 | 0.36 | 60.03 |
| 0.8 | 409.05 | 0.12 | 24.25 | 177.39 | 0.8 | 71.97 | 328.87 | 0.4 | 51.51 |
| 0.95 | 330.51 | 0.14 | 24.25 | 130.76 | 1.38 | 71.97 | 291.05 | 0.43 | 45.11 |

properties result mainly from the droplet surface tension (the Kelvin effect). The predictions with particles containing strong surfactant depend on the modeling method and emphasize the need for validation of the different bulk–surface partitioning

models across a wide range of particle mixtures and conditions. Uncertainty in predicting the critical properties of activating droplets creates uncertainty in generalizations for atmospheric processes such as estimating the number of cloud droplets, cloud albedo, and radiative forcing.

The inter-model variation for common aerosol systems comprising NaC14 and analogous predictions for particles containing malonic, succinic, or glutaric acid (Vepsäläinen et al., 2022) differ. In all cases investigated so far, the differences between

the predictions of the different models increase with the mass fraction of the surface active compound in the particles. The predictions of cloud droplet activation for a given aerosol system containing either moderately surface active compounds or strong surfactants depend on the modeling method. Details of which models mutually agree or disagree in predictions of critical droplet properties for identical initial particles vary between the strong and moderate surfactants studied. This highlights how models must be validated for a range of surface active aerosol systems and surfactant properties, as well as atmospheric

conditions, before their broad applicability for atmospheric modeling can be established. Otherwise significant errors could be introduced in modeling larger scale atmospheric processes through the generalization of Köhler predictions created for only a given few systems or conditions.

*Author contributions.* SV performed the simulations and the analysis of model results with assistance from SMC and NLP. SV wrote the original manuscript draft and made the visualizations with NLP and assistance from SMC. NLP conceived the project and methodology, was

responsible for supervision and project management, and secured funding for the work.



*Competing interests.* There are no conflicts to declare.

*Acknowledgements.* We thank Chris Ruehl and Chris Cappa for providing and discussing their codes of the compressed film model. We also thank Andreas Zuend for providing the surface coverage and surface tension calculation routine used for the partial organic film model. This project has received funding from the European Research Council (ERC) under the European Union's Horizon 2020 research and innovation programme, Project SURFACE (grant agreement no. 717022). The authors also gratefully acknowledge the financial contribution from the Academy of Finland (grant nos. 308238, 314175 and 335649).




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
