# Peer review of "Comparison of six approaches to predicting droplet activation of surface active aerosol – Part 2: strong surfactants"

_EGUsphere, 2022_

## Author Comment (AC1)

**Author response to reviewers' comments**

Sampo Vepsäläinen[1], Silvia M. Calderón[2], and Nønne L. Prisle[3]

[1]Nano and Molecular Systems Research Unit, University of Oulu, P.O. Box 3000, FI-90014, Oulu, Finland
[2]Finnish Meteorological Institute, P.O. Box 1627, FI-70211, Kuopio, Finland
[3]Center for Atmospheric Research, University of Oulu, P.O. Box 4500, FI-90014, Oulu, Finland

**Correspondence:** Nønne L.Prisle (nonne.prisle@oulu.fi)

We thank both reviewers for their careful revision of our manuscript and constructive comments. Below we provide our responses to specific comments in a point-by-point manner. The reviewers' comments are reproduced in *italics*, our responses in blue and quotes from the revised manuscript in **red bold** font. We have split some of the longer reviewer comments into specific points, so that the connection between comments and responses is easier to follow.

For the revised manuscript, we have also updated the calculations, figures and tables after discovering some minor mistakes in the model setup. These have minimal impact on the results, and do not affect our interpretation of the results or the overall conclusions, compared to the first version of the manuscript. The most noticeable change occurs for calculations with the bulk solution model, where we found that the direction of the CMC constraint in the calculation of water activity was reversed, leading to somewhat underestimated $a_w$ values, in the original calculations. The bulk solution model is included in this work as a reference model for the different partitioning models, which are the main focus of our comparison.

**1 Referee 2**

*Vepsäläinen and co-authors report the results of a modeling study in which six models of surfactant action in cloud condensation nucleation are explored for strong surfactants. The models are based on surfactant properties measured for bulk, and each model accounts for the aerosol phase in a different way (including a model in which the bulk properties are used as-is). Each model has a different effect on the activation of CCN, as demonstrated by Kohler theory. The models are compared, and differences are discussed. Though the manuscript is well-written and well-organized, I find that there are some shortcomings of the paper that need to be addressed before publication. In its current form, the manuscript would be better suited as a technical note.*

**1.1**

*For publication as a research article, I recommend adding a discussion of the how these model differences propagate into uncertainty in cloud droplet number or a similar impact on the aerosol-cloud-climate system.*

We have added a new Results and Discussion Section 3.4 to the revised manuscript. The new section explores the relative change in cloud droplet number concentration with the predictions of the various surface activity models, compared to those of a classical Köhler model, where effects of surface activity are disregarded. We also added new discussion on the potential

25 larger-scale implications of the differences between the model predictions. More details on these results are given below in Section 1.3.4 of this response.

**1.2**

*Further, I find that there are a few assumptions in the model that should be discussed a little more. There is no Discussion section – perhaps there should be. If these concerns can be addressed, I think the work would be suitable for prompt publication and*
30 *would be of interest to the community.*

We emphasise that this manuscript presents a comparison of state-of-the-art models available in the literature and currently in use by the community. Our aim is to compare the models as they are presented and currently used and we do not attempt to improve the models in the present work. For exhaustive documentation of each model and its applications, we refer to the original publications. It is outside the scope and purpose of the present work to evaluate or validate the underlying assumptions
35 of each model. Our purpose is to compare the predictions of the various bulk–surface partitioning models for the same aerosol systems and conditions, to assess the robustness of our understanding of bulk–surface partitioning effects in cloud droplet activation, as represented by the mutual agreement of these existing frameworks, which are currently used to predict or interpret the behavior of surface active organic aerosol in the laboratory or in the atmosphere. Rather than a separate Discussion section, the scientific discussion of our results is presented as part of Section 3: Results and Discussion of both the preprint and revised
40 manuscript.

**1.3 Specific comments**

**1.3.1**

*Line 123-125. how confident are the authors in the assumption of volume additivity? Granted, this has always been the assumption for Kohler theory. Many mixtures do not mix with volume additivity in the bulk – how might this affect the Kohler*
45 *curves shown here? Is density of particles a function of size?*

Assumptions of volume additivity are common in Köhler calculations, as also noted by the reviewer, and is indeed assumed in connection with several of the models employed in this work, as detailed in the respective publications presenting each of the models (e.g. Prisle et al., 2010, 2011; Ruehl et al., 2016; Ovadnevaite et al., 2017, and references therein). In our calculations, the density of the aqueous droplet solution, where it is a required input for the model, is a function of composition
50 and is therefore affected by the size of the droplet through dilution as the water content in the droplet increases. The growing droplets discussed in the present work are dilute aqueous solutions, especially near the point of cloud droplet activation, and here the assumption of volume additivity is reasonable. The densities of dilute aqueous solutions comprising the considered surface active organics are furthermore close to the density of water (Calderón and Prisle, 2021). Prisle et al. (2008) tested the assumption of additive volumes of the dry particle components and the condensed water in the droplets by comparing Köhler
55 calculations for sodium octanoate particles with a Gibbs partitioning model where the total amount of water in droplets was

obtained from either the relation

$$(V_{\text{droplet}} - V_{\text{particle}})/\bar{v}_{\text{w}}, \tag{1}$$

where $\bar{v}_{\text{w}} = M_{\text{w}}/\rho_{\text{w}}$, or from

$$(\rho_{\text{solution}}V_{\text{droplet}} - \rho_{\text{particle}}V_{\text{particle}})/M_{\text{w}}. \tag{2}$$

60   The calculated values of $SS_c$ were very close for the two approaches to calculation of droplet density, with a relative difference of only 0.1 % for dry particles with diameter of 40 nm, and less than 0.1 % for particles with a dry diameter of 100 nm. We therefore consider the assumption of volume additivity to affect the relevant models in a consistent and fairly even manner and not to impact the overall conclusions or interpretation of the results.

The sentence in question has been clarified:

65   **In addition, the position of the Gibbs diving surface is selected so that the droplet bulk-phase volume equals the total equimolar droplet volume, and mass conservation ($n_j^{\text{T}} = n_j^{\text{S}} + n_j^{\text{B}}$) is assumed for all components in the droplet (e.g. Prisle et al., 2010).**

We again emphasize that our purpose is not to evaluate or validate the underlying assumptions of the different models of cloud droplet surface activity. Our aim is to compare the predictions of the different bulk–surface partitioning models for the
70   same systems and conditions, to assess the robustness of the current understanding in the community of the effects of aerosol surface activity on hygroscopic growth and CCN activation.

**1.3.2**

*Line 262. it seems like the CMC will depend rather strongly on the NaCl content. Please comment on this – can the uncertainty here be constrained?*

75   We have briefly mentioned the effects of NaCl on the CMC of the surfactant in Section 2.8 in the preprint. To account for this effect, it is necessary to use complex models with parameters obtained from experiments that are scarce or nonexistent for the ternary systems in question. As explained above, our main purpose is not to evaluate the surface activity models used in the present work, or to provide constraints on the models or model parameters. Instead, our aim is to compare the predictions of the different bulk–surface partitioning models for the same droplet systems and conditions, to assess their robustness in capturing
80   the properties of a given droplet system comprising surface active organic aerosol components.

We have expanded the discussion in Section 2.9 of the revised manuscript to further clarify the effects of NaCl on the surfactant CMC:

**In ternary water–surfactant–inorganic salt solutions, both the surfactant CMC and solution surface tension at the CMC can vary with inorganic salt concentration and may be further affected if the inorganic salt and an ionic surfactant**
85   **share a common counterion. Accounting for these effects requires complex modeling (e.g., Kralchevsky et al., 1999; Calderón et al., 2020) with parameters obtained from experiments. The surfactant CMC often decreases with increasing inorganic salt concentration, compared to a binary water–surfactant solution (Calderón and Prisle, 2021). However, for**

**most ternary surfactant solutions, and in particular for atmospheric surfactants, the relevant interaction parameters are not known and readily accessible with existing experimental techniques.**

The discussion of the results around line 262 (of the preprint) has also been extended:

**Previous Köhler calculations have shown that concentrations of surface active fatty acid salts were below the respective CMCs at the point of cloud droplet activation (Prisle et al., 2008, 2010). However, the CMC for $NaC_{14}$ was in this work estimated from the surface tension data of Wen et al. (2000), whereas Prisle et al. (2008) used the data of Campbell and Lakshminarayanan (1965). Experimentally determined CMC values can vary significantly for a given solution, depending on the measurement technique used (Álvarez Silva et al., 2010).**

**1.3.3**

*Line 272-273. even if these acids were to partition very strongly, would the surface tension be suppressed? Does the mixture's surface tension depend on surface concentration necessarily? For example, pure liquids also can have a suppressed surface tension relative to water.*

The question pertains to the fundamental reason why bulk–surface partitioning models are needed. Detailed explanations for this point are given in the introductions of Lin et al. (2020) and Prisle (2021). We briefly highlight the main points here, together with clarifications made in the revised manuscript. The surface tensions of many pure, potentially sub-cooled, organic acids can be significantly lower than that of pure water (e.g. Hyvärinen et al., 2006; Vanhanen et al., 2008). This is the basis for the ability of these compounds to lower the surface tension of aqueous mixtures. When these compounds have enhanced surface concentrations, compared to an isotropic mixture, due to surface activity, the ability to reduce the surface tension of the aqueous solution is further enhanced. Bulk–surface partitioning in finite-sized droplets alters the surface enhancement at a given total concentration of the surface active solute, compared to a macroscopic solution, due to the simultaneous bulk-phase depletion. This effect is taken into account with the bulk–surface partitioning model. The model captures the bulk–surface partitioning at a given droplet state, and therefore the surface enhancement of the organic and resulting surface tension depression, as a function of either the bulk or surface concentration. Because experimental surface tension relations are far more readily obtained in terms of bulk concentration (e.g. Prisle et al., 2008, 2010; Malila and Prisle, 2018; Lin et al., 2018, 2020; Bzdek et al., 2020; Prisle, 2021), we typically use these relations to describe the surface composition and surface tension of the droplets. Surface tension is typically a decreasing function of surfactant bulk phase concentration. In microscopic droplets that contain a finite amount of solute, including the surface active compound, the surface tension depression may be dampened when the droplet bulk phase is strongly depleted from organic bulk–surface partitioning (Prisle et al., 2010; Bzdek et al., 2020; Prisle, 2021). In these droplets, also the surface concentration is reduced, compared to a macroscopic solution.

We have revised the following part in the Introduction of the original preprint:

**In microscopic droplets, the partitioning of surfactant mass between the droplet bulk and surface has been predicted to significantly alter the bulk composition, due to the finite total amount of surfactant solute contained in such small droplets, and therefore affect composition-dependent droplet properties, such as the surface tension (e.g., Prisle et al., 2010; Lin et al., 2018, 2020; Bzdek et al., 2020; Prisle, 2021). The surface tension of a solution can be described in terms**

of either the surface or bulk composition, which for a given surface-active substance are related via the equilibrium bulk-to-surface concentration gradient. The relation between surface tension and surface-specific composition is often unknown (Prisle et al., 2012a; Werner et al., 2014, 2018) and experimentally based surface tension relations are therefore typically expressed in terms of the solution bulk composition. The surface tensions of aqueous solutions are often decreasing functions of the concentration of surface active compounds (e.g., Wen et al., 2000; Hyvärinen et al., 2006; Vanhanen et al., 2008; Bzdek et al., 2020). In a macroscopic solution with a total composition identical to that of a microscopic droplet, the bulk phase contains a sufficiently large amount of surfactant that bulk–surface partitioning has negligible effect on the composition of the bulk. The surface tension is therefore readily described in terms of the total solution composition. For microscopic droplets, the change in bulk composition due to bulk–surface partitioning can be estimated using a *bulk–surface partitioning model*, which accounts for the material partitioned to the droplet surface to correct the bulk composition of a droplet. This allows the use of macroscopic solution properties for microscopic droplets (Prisle et al., 2010; Lin et al., 2018, 2020; Bzdek et al., 2020; Prisle, 2021).

We have also expanded the discussion in the Results and Discussion around lines 272-273 of the preprint:

For particles comprising malonic, succinic, or glutaric acid mixed with ammonium sulfate (Vepsäläinen et al., 2022), the bulk solution model did not lead to significantly lower predicted critical supersaturations than the bulk—surface partitioning models, due to the moderate surface activity of these acids. The monolayer and Gibbs partitioning models predicted only moderate extent of surface partitioning for the three acids and consequently also bulk-phase compositions similar to the bulk solution model. Therefore, these two partitioning models generally predicted similar Köhler curves and droplet surface tensions to the bulk solution model (Vepsäläinen et al., 2022).

**1.3.4**

*Line 445. can the authors make an estimate of the change in uncertainty in, e.g., cloud droplet number concentration, with the change in surface tension model? This is missing from the discussion.*

The different models that we use are not simply different surface tension models, but rather bulk–surface partitioning models which allow estimations of significantly altered bulk-phase concentrations for microscopic droplets, due to partitioning. These partitioning models are different, but together represent the current state-of-the-art of bulk–surface partitioning models available in the literature. In particular, the partitioning models employ various methods or assumptions to determine the droplet surface tension. The bulk solution model is included to the comparison as a reference that accounts for reduced surface tension, but does not take the effects of bulk–surface partitioning into account in estimating the surface tension of droplets. A more detailed explanation of why partitioning models are needed is provided above in Section 1.3.3 of this response.

We have added new results (Section 3.4 of the revised manuscript) showing the relative change in cloud droplet number concentration by considering organic aerosol surface activity with the different bulk–surface partitioning models and the bulk solution model, compared to a classical Köhler model which does not explicitly consider any effects of surface activity. The particle size range for these additional results has been extended to $D_\mathrm{p} = 50 - 200$ nm following the comment by reviewer 3 below in Section 2.5 of this response. Classical Köhler models are by far the most frequently used in larger-scale simulations,

with only a few exceptions (e.g., Prisle et al., 2012b; Lowe et al., 2019). For completeness, we have now also included predictions with the classical Köhler model in the Results and Discussion and a short description of the classical Köhler model has been added in Section 2.8 of the revised manuscript:

**In the classical Köhler model, the specific effects of surface activity are disregarded, such that the droplet surface tension is assumed to be constant and equal to that of pure water. The water activity is calculated as a corrected mole fraction corresponding to the total composition of the droplet.**

The following explanation about estimating the relative change in the cloud droplet number concentration was added as a new Section 2.10 of the revised manuscript:

**To obtain an estimate of the relative change in cloud droplet number concentration caused by the differences in critical supersaturation predicted with the different partitioning models, as well as the bulk solution model, we follow the method outlined by Bzdek et al. (2020) and references therein. The cloud droplet number concentration $N$ is assumed to depend on the supersaturation SS as $N \propto \mathrm{SS}^k$, where $k \approx 0.5$ (Facchini et al., 1999), and the relative change in cloud droplet number concentration is calculated as**

$$\frac{\Delta N}{N} = \frac{(\mathrm{SS}_c^0)^k - (\mathrm{SS}_c)^k}{(\mathrm{SS}_c)^k}. \tag{3}$$

**Here, $\mathrm{SS}_c^0$ is the critical supersaturation for droplet activation in the reference case and $\mathrm{SS}_c$ is the critical supersaturation predicted with the different partitioning models and the bulk solution model. The reference case for $\Delta N N^{-1}$ due to effects of organic aerosol surface activity is considered to be the classical Köhler model, which is used in most larger-scale simulations, with a few exceptions (e.g., Prisle et al., 2012b; Lowe et al., 2019).**

The new results showing the relative change in the cloud droplet number concentration have been added as Section 3.4 of the revised manuscript:

**Figure 4 shows the relative change in cloud droplet number concentration ($\Delta N N^{-1}$, Eq. (3)) for $D_p = 50 - 200$ nm, predicted with the different bulk–surface partitioning models and the bulk solution model to account for aerosol surface activity. The relative change is calculated with respect to predictions of the classical Köhler model. The magnitude of $\Delta N N^{-1}$ illustrates how variation in critical supersaturations predicted with the different models of surface activity translates into uncertainty in estimates of the cloud droplet number concentration based on Köhler theory.**

**In Fig. 4, the monolayer, simple partitioning, compressed film, and partial film models all predict very little variation in $\Delta N N^{-1}$ with $D_p$, compared to the classical Köhler model predictions for each surfactant mass fraction ($w_{p,sft}$) in the particles. The differences between the model predictions of $\Delta N N^{-1}$ increase with $w_{p,sft}$, as also seen with the $\mathrm{SS}_c$ in Fig. 1. At $w_{p,sft}$ = 0.2, the predictions with the monolayer, simple partitioning, compressed film, and partial film models each yield $\Delta N N^{-1}$ within $\pm 4$ % of the classical Köhler model. For $w_{p,sft}$ = 0.95, we predict 29 % and 28 % more cloud droplets with the monolayer and partial film models, respectively, than with the classical Köhler model, on average over the investigated particle size range. This is in reasonable agreement with results of Lowe et al. (2019), who used a partial film partitioning model combined with a cloud parcel model to predict increased cloud droplet number concentrations compared to a classical Köhler model: an increase of 13 % for marine and 26 % for continental boreal**

aerosol populations. For $w_{\mathrm{p,sft}}$ = 0.95, we however predict a 28 % decrease in in $\Delta NN^{-1}$ with the compressed film model and a decrease of 35 % with the simple model, compared to the classical Köhler model.

With the Gibbs model, we here predict an increase of $2-21$ % in $\Delta NN^{-1}$ on average over the particle size range for $w_{\mathrm{p,sft}} = 0.2-0.95$. The bulk solution model predicts the lowest $\mathrm{SS_c}$ for the cases studied, resulting in the largest $\Delta NN^{-1}$. This is similar to the results of Prisle et al. (2012b), who implemented surfactant effects in the global circulation model (GCM) ECHAM5.5–HAM2. The bulk solution model predictions of $\Delta NN^{-1}$ are here seen to decrease ($w_{\mathrm{p,sft}}$ = 0.2 and 0.5), increase ($w_{\mathrm{p,sft}}$ = 0.95), or change non-monotonically ($w_{\mathrm{p,sft}}$ = 0.8) as a function of $D_{\mathrm{p}}$. This variation is due to the constraint on the droplet water activity imposed by the CMC, as well as the critical point moving between the two maxima of the Köhler curve located at droplet sizes before and after the droplet surface tension increases from the constant value at the CMC (see Figs. 1 and 2). On average over the particle size range, the bulk solution model predicts increases of $55-99$ % in $\Delta NN^{-1}$ for the different surfactant mass fractions in the particles. However, these very large $\Delta NN^{-1}$ predicted with the bulk solution model should be considered with some caution, as the large relative changes in $\mathrm{SS_c}$ may conflict with the assumption of simple exponential dependence in Eq. (3). The partitioning models are considered to give the more comprehensive and realistic representations of the behavior of the droplet system, whereas the bulk solution model has been included in this comparison for reference.

The present results support previous work, where the surface activity of organic aerosol and its representation in calculations of cloud droplet activation were found to have the potential to significantly influence global-scale predictions, at least for specific regions (Prisle et al., 2012b). Increased cloud droplet number concentrations would ultimately have a negative radiative effect and, therefore, a cooling effect on the climate. Decreased cloud droplet number concentrations would conversely lead to a warming effect. Facchini et al. (1999) estimated that a 20 % increase in the cloud droplet number concentration corresponds to a change in cloud radiative forcing of $-1\ \mathrm{Wm}^{-2}$, in very good agreement with the full GCM predictions of Prisle et al. (2012b) for similar conditions. The variation in predicted $\Delta NN^{-1}$ between the different models for high mass fractions of $\mathrm{NaC_{14}}$ in the present work suggests that the representation of surfactant effects could translate to significant uncertainty in larger-scale predictions of cloud radiative effects for regions where strongly surface active aerosol are prevalent. Both significant warming or cooling effects, compared to the predictions of a classical Köhler model, could result, depending on the specific partitioning model used. Therefore, bulk–surface partitioning effects in Köhler calculations should be established for a wide range of conditions and aerosol types relevant to the atmosphere. The agreement between the models should be robust before conclusions of the effects can be generalized to other types of aerosol systems, just as conclusions regarding bulk–surface partitioning effects for any one system should be interpreted with some caution.

The Conclusions around line 445 of the preprint have been edited to reflect these new additional results:

Our estimations show that the relative change in cloud droplet number concentration for strongly surface active aerosol can vary significantly between the predictions of the different models. Predictions of $\Delta NN^{-1}$ compared to a classical Köhler model vary from -35 % to 29 % between the partitioning models, corresponding to both warming and cooling climate effects. These differences represent a significant uncertainty in estimating the cloud radiative effects for regions

225 **where strongly surface active aerosol are prevalent. Therefore, conclusions based on any one surface activity model should not immediately be broadly generalized.**

**1.4 Technical corrections**

*Line 3. depleting the droplet bulk? This is a little ambiguous*

Clarification added:

230 **...depleting the droplet bulk of the surfactant.**

*Line 83. the notation of NaC14 is very confusing, as it seems to imply NaCl_4. Maybe the "14" could be a subscript?*

NaC14 has been changed to $NaC_{14}$.

*Line 90. extra period after "Table 1"*

The extra period has been deleted.

235 *Line 210. Extra period between sentence and citation*

The period has been moved after the citation.

*Line 261-262. this does not seem like a numerical artifact – I think the sentence needs to be clarified*

The sentence has been clarified:

**However, the critical points predicted with the Gibbs model correspond to droplet sizes immediately after the droplet**
240 **surface tension increases from the value at the CMC (see Fig. 2).**

**2 Referee 3**

**2.1**

*This paper is written largely as a repetition of Part 1: S. Vepsäläinen et al.: "Comparison of six approaches to predicting droplet activation of surface active aerosol – Part 1: moderately surface active organics". The only differences I see are (1)*
245 *that the six approaches are more completely described in Part 1, and (2) Parts 1 and the submission treat different aerosol: NaCl/Sodium myristate (NaC14) in the submission versus (NH4)2 SO4/Malonic acid in Part 1.*

Our overall purpose with these works is to compare Köhler predictions of cloud droplet activation with different bulk–surface partitioning models for common systems and conditions, to assess the robustness of these-state-of-the-art frameworks currently in use by the community. This inter-model comparison should be carried out for a variety of different types of surface
250 active aerosols, for which the relative behavior of these models, as well as the behavior of each individual model, may differ significantly, due to variations in the expression of surface active properties. In Vepsäläinen et al. (2022), we compared surface activity models for common moderately surface active aerosol components, represented by malonic, succinic, and glutaric acids. We focused the discussion on the results for malonic acid, due to the similarity between the results for these three organic acids, whereas results for succinic and glutaric acid aerosols are presented in the supplement of Vepsäläinen et al.
255 (2022). However, it is not given that the inter-comparison between the different surface activity models can immediately be

generalized to other surface active aerosol systems or conditions, where the effects of surface activity on cloud droplet activation may be expressed differently. The surface adsorption properties differ between strong surfactants and moderately surface active compounds. In this work, we therefore investigate whether the results of the inter-comparison between the different surface activity models in Köhler theory for common systems of moderate surface active compounds also apply for strongly surface active aerosols.

We find both similarities and significant differences between the partitioning model inter-comparison for the strongly surface active aerosols studied in the present work and the moderately surface active compounds previously studied by Vepsäläinen et al. (2022). Therefore, the robustness of the partitioning model predictions cannot be assessed from comparing predictions for only a single or few surface active aerosol systems. Conclusions made for any one type of aerosol cannot be generalized for different surface active aerosols without investigating a variety of systems.

**2.2**

*For example Part 1 provides a conceptual figure of the different models that is not present in the submission. To get such detail the reader has to go back and forth between the two papers (Part 1 and submission ) anyway – so why not, in the submitted paper, save journal space and simply refer to Part 1 for the six models and the theory already described there throughout?*

We have carefully considered how to strike the balance between the detailed descriptions included in Vepsäläinen et al. (2022) and providing necessary information readily available for the present work with a minimum of repetition. We prefer to provide relevant information on the key concepts in the present manuscript. In response to the reviewer's comments about saving journal space, we have attempted to further condense the descriptions in the current work. However, there are differences between the current manuscript and Vepsäläinen et al. (2022) that make further condensation of the model descriptions difficult. In particular, the explanation of relevant details, such as the model-specific parameters and parameterizations for the physico-chemical properties of the droplet solution, etc., require sufficient context. The most significantly shortened descriptions are for the general Köhler calculations (Section 2.1 of the revised manuscript), and the partial organic film model (Section 2.5 of the revised manuscript). The other model descriptions have only been slightly edited.

During the revisions, we also added a classical Köhler model to the comparison, to elaborate on potential larger-scale atmospheric effects that could be caused by the differences between the predictions of the different surface activity models. The details can be found above in Section 1.3.4 of this response. The classical Köhler model is briefly described together with the other models in Section 2.8 of the revised manuscript.

**2.3**

*The authors refer frequently to Part 1, but I find these comparisons to be more descriptive than insightful. It would be useful to provide the reader with a broader understanding/overview of the effects of moderate versus strong surfactants learned from the two papers as a whole. Part 1 succeeds better in this respect as well, with a conclusions section that makes broad connections of that work to cloud microphysics, fluctuations in supersaturation, and other large-scale effects. The submitted manuscript gives neither discussion nor conclusions, which is suprising in that there are now two papers in the series to draw from.*

Our purpose with both the present work and previous work of Vepsäläinen et al. (2022) is not to assess or compare the effects of strong and moderately surface active aerosols in cloud droplet activation. We are comparing the predictions of different bulk–surface partitioning models for the same aerosol systems and conditions, in order to assess the robustness of these existing state-of-the-art models currently in use, for different types of surface active aerosol systems. We perform this inter-model comparison for different types of surface active aerosols, where the expression of surface activity and the relative ability of the different surface activity models to capture these properties may differ. The robustness of our understanding of the comprehensive effects of surface active aerosol on cloud droplet activation that is revealed by the comparison of these existing models is investigated for 1) the specific surface active aerosol systems treated in each of the manuscripts, and 2) between aerosol systems with different surface active properties. We have further emphasized and clarified these points and the purpose of this work in the Results and Discussion, as well as the Conclusions of the revised manuscript.

To elaborate on the connections of the present partitioning inter-model comparison to larger-scale phenomena, we have added new results and discussion (Section 3.4 of the revised manuscript) showing the relative change in cloud droplet number concentration from Köhler predictions with the different partitioning models and the bulk solution model. The relative change is calculated with respect to a classical Köhler model that does not include effects of surface activity. The Conclusions have also been edited to reflect these new results. More details have been given above in response to the specific comment from reviewer 2 in Section 1.3.4 of this response. Additionally, the discussion related to the calculated Köhler curves has been extended in the Results and Discussion Section 3.1:

**The predictions of $SS_c$ for $NaC_{14}$ particles with $D_p = 50$ nm are between $0.09 - 1.62$ % (see Table 2). Average supersaturations in low-level clouds range from 0.1 % to 0.4 % (e.g. Politovich and Cooper, 1988) but higher supersaturations of 0.7 % to 1.3 % can be reached during turbulent fluctuations of temperature and water vapor concentration (e.g. Siebert and Shaw, 2017). Only the simple partitioning, compressed film, and classical Köhler models predict $SS_c$ above 0.7 % at high surfactant mass fractions, while the other models predict considerably lower $SS_c$ values (Table 2). Therefore, it is difficult to assess whether activation of droplets similar to those studied here occurs during turbulent fluctuations of temperature and water vapor concentration.**

A more general comparison to the results of Vepsäläinen et al. (2022) has also been added to the same paragraph:

**In contrast, the range of $\Delta SS_c$ predicted for malonic acid droplets by Vepsäläinen et al. (2022) are fairly similar to the $\Delta SS_c$ presented here for $NaC_{14}$, whereas predictions of $SS_c$ for moderately surface active organic aerosol were generally higher than those predicted here for strongly surface active aerosol. Therefore, Vepsäläinen et al. (2022) considered the activation of particles containing high mass fractions of moderately surface active compounds to occur mostly during turbulent fluctuations of temperature and water vapor concentration in low-level clouds.**

We have also added more emphasis on the inter-model comparison between the strongly and moderately surface active aerosols studied between the present manuscript and Vepsäläinen et al. (2022), both in the Results and Discussion and Conclusions of the revised manuscript. In the Results and Discussion, we have added the following at the end of Section 3.1:

**The inter-comparison of partitioning models show several difference between strongly and moderately surface active aerosols (Vepsäläinen et al., 2022). This suggests that the robustness of our understanding of effects of aerosol surface**

**activity on cloud droplet formation captured by the different partitioning models cannot be assessed based on a single or few surface active aerosol systems. We found the mutual agreement between the different partitioning models to vary with the surface active properties of the aerosol system. Therefore, conclusions about which surface activity models are mutually consistent cannot immediately be generalized from one aerosol system, such as moderately surface active aerosol (Vepsäläinen et al., 2022), to different types of surface active aerosols. As a consequence, asessements of bulk–surface partitioning effects in Köhler calculations should not be based on predictions for a single or few systems, but rather for a wide range of conditions and aerosol types relevant to the atmosphere, to ensure that our understanding of the effects of surface activity as captured by the various models is robust.**

More emphasis on the inter-model comparison between different aerosol systems has been added also in the Conclusions:

**The inter-model variation differs significantly for aerosol systems comprising strongly and moderately (Vepsäläinen et al., 2022) surface active compounds. For the aerosol types and conditions investigated so far, the inter-model differences increase with the mass fraction of surface active compound in the particles. The predictions of cloud droplet activation for a given aerosol system comprising either moderately or strongly surface active compounds depend on the model for surface activity, but the mutual agreement between models for identical aerosols varies with aerosol surface activity. Therefore, conclusions about the robustness of our understanding of the effects of aerosol surface activity, as captured by the inter-model variation, cannot immediately be generalized for different surface active organic aerosol systems. This highlights how aerosol surface activity models must be validated for a range of surface active aerosol types and ambient conditions, before establishing their broad applicability in atmospheric modeling. Bulk–surface partitioning can significantly affect predictions of cloud droplet activation, but the exact impact as predicted with any model should be assessed with caution, as the ability to capture the effects of surface activity varies significantly with the partitioning model used and aerosols of different surface activity. Generalization of Köhler predictions for only a few surface active aerosol systems and conditions could introduce significant errors in modeling larger-scale atmospheric processes.**

**2.4**

*In summary, the authors should shorten their paper given the repetitive overlap with Part 1 noted in the first paragraph of this review. Moreover, the authors should end their paper by providing general discussion along the lines described in the second paragraph of this review – if for no other reason than to contextualize their point-by-point and figure-by-figure comparisons made throughout the paper.*

The reviewers' comments concerning 1) relation of the current manuscript to Vepsäläinen et al. (2022) and 2) shortening of the manuscript are both discussed above in Section 2.1 of this response. The Results and Discussion of the revised manuscript now also 3) contains a broader discussion about the potential larger-scale effects caused by differences between the predictions of the various models for aerosols containing $NaC_{14}$. Furthermore, 4) more focus has been given to clarify the importance of the inter-model comparison between the strongly and moderately surface active aerosols studied between the current work and Vepsäläinen et al. (2022). The details relating to points 3) and 4) can be found in Section 2.3 of this response.

**2.5**

*Finally, the authors should consider, perhaps in a separate figure, expanding the range of particle size, now limited to the*
*single size of Dp = 50nm. The most important location along any Kohler curve is the critical point, or maximum, marking*
*the threshold for cloud droplet activation. Showing a locus of these points as particle size is varied for one or more of the six*
*models would add this new dimension.*

We have extended the calculations for a wider particle size range of $D_{\mathrm{p}} = 50 - 200$ nm. New results are presented in Section
3.4, where we present the relative change in the cloud droplet number concentration in response to feedback from reviewer 2
(Section 1.3.4 of this response). We have also included additional results for the same extended dry particle size range in the
supplementary material and added the following at the beginning of the Results section:

**Furthermore, predictions of critical supersaturation and diameter for the dry particle size range of $D_{\mathrm{p}} = 50 - 200$ nm**
**are presented in Section S1.4 of the Supplement. The inter-comparison between the different surface activity models**
**for a given particle mass fraction of NaC$_{14}$ remains largely the same for all particle sizes. The differences in predicted**
**SS$_{\mathrm{c}}$ between the models are larger for small particles, and therefore here we focus on the results for $D_{\mathrm{p}} = 50$ nm.**

**References**

[revised manuscript text omitted]

---

## Author Response (AR1)

**Author response to reviewers' comments**

Sampo Vepsäläinen[1], Silvia M. Calderón[2], and Nønne L. Prisle[3]

[1]Nano and Molecular Systems Research Unit, University of Oulu, P.O. Box 3000, FI-90014, Oulu, Finland
[2]Finnish Meteorological Institute, P.O. Box 1627, FI-70211, Kuopio, Finland
[3]Center for Atmospheric Research, University of Oulu, P.O. Box 4500, FI-90014, Oulu, Finland

**Correspondence:** Nønne L.Prisle (nonne.prisle@oulu.fi)

We thank both reviewers for their careful revision of our manuscript and constructive comments. Below we provide our responses to specific comments in a point-by-point manner. The reviewers' comments are reproduced in *italics*, our responses in blue, and quotes from the revised manuscript in **red bold** font. We have split some of the longer reviewer comments into specific points, so that the connection between comments and responses is easier to follow.

5   For the revised manuscript, we have also updated the calculations, figures and tables after discovering some minor mistakes in the model setup. These have minimal impact on the results, and do not affect our interpretation of the results or the overall conclusions, compared to the first version of the manuscript. The most noticeable change occurs for calculations with the bulk solution model, where we found that the direction of the CMC constraint in the calculation of water activity was reversed, leading to somewhat underestimated $a_\mathrm{w}$ values, in the original calculations. The bulk solution model is included in this work
10   mainly for comparison with the different partitioning models, which are the main focus of our comparison.

**1   Referee 2**

*Vepsäläinen and co-authors report the results of a modeling study in which six models of surfactant action in cloud conden-sation nucleation are explored for strong surfactants. The models are based on surfactant properties measured for bulk, and each model accounts for the aerosol phase in a different way (including a model in which the bulk properties are used as-is).*
15   *Each model has a different effect on the activation of CCN, as demonstrated by Kohler theory. The models are compared, and differences are discussed. Though the manuscript is well-written and well-organized, I find that there are some shortcomings of the paper that need to be addressed before publication. In its current form, the manuscript would be better suited as a technical note.*

**1.1**

20   *For publication as a research article, I recommend adding a discussion of the how these model differences propagate into uncertainty in cloud droplet number or a similar impact on the aerosol-cloud-climate system.*

We have added a new Results and Discussion Section 3.4 to the revised manuscript. The new section explores the relative change in cloud droplet number concentration with the predictions of the various surface activity models, compared to those of a classical Köhler model, where effects of surface activity are disregarded. We also added new discussion in Section 3.4 of the

25 revised manuscript on the potential larger-scale implications of the differences between the model predictions. More details on these results are given below in Section 1.3.4 of this response.

**1.2**

*Further, I find that there are a few assumptions in the model that should be discussed a little more. There is no Discussion section – perhaps there should be. If these concerns can be addressed, I think the work would be suitable for prompt publication and*
30 *would be of interest to the community.*

We emphasise that this manuscript presents a comparison of state-of-the-art models available in the literature and currently in use by the community. Our aim is to compare the models as they are presented and currently used and we do not attempt to improve the models in the present work. For exhaustive documentation of each model and its applications, we refer to the original publications. It is outside the scope and purpose of the present work to evaluate or validate the underlying assumptions
35 of each model. Our purpose is to compare the predictions of the various bulk–surface partitioning models for the same aerosol systems and conditions, to assess the robustness of our understanding of bulk–surface partitioning effects in cloud droplet activation, as represented by the mutual agreement of these existing frameworks, which are currently used to predict or interpret the behavior of surface active organic aerosol in the laboratory or in the atmosphere. Rather than a separate Discussion section, the scientific discussion of our results is presented as part of Section 3: Results and Discussion of both the preprint and revised
40 manuscript.

**1.3 Specific comments**

**1.3.1**

*Line 123-125. how confident are the authors in the assumption of volume additivity? Granted, this has always been the assumption for Kohler theory. Many mixtures do not mix with volume additivity in the bulk – how might this affect the Kohler*
45 *curves shown here? Is density of particles a function of size?*

Assumptions of volume additivity are common in Köhler calculations, as also noted by the reviewer, and is indeed assumed in connection with several of the models employed in this work, as detailed in the respective publications presenting each of the models (e.g. Prisle et al., 2010, 2011; Ruehl et al., 2016; Ovadnevaite et al., 2017, and references therein). In our calculations, the density of the aqueous droplet solution, where it is a required input for the model, is a function of composition
50 and is therefore affected by the size of the droplet through dilution as the water content in the droplet increases. The growing droplets discussed in the present work are dilute aqueous solutions, especially near the point of cloud droplet activation, and here the assumption of volume additivity is reasonable. The densities of dilute aqueous solutions comprising the considered surface active organics are furthermore close to the density of water (Calderón and Prisle, 2021). Prisle et al. (2008) tested the assumption of additive volumes of the dry particle components and the condensed water in the droplets by comparing Köhler
55 calculations for sodium octanoate particles with a Gibbs partitioning model where the total amount of water in droplets was

obtained from either the relation

$$(V_{\text{droplet}} - V_{\text{particle}})/\bar{v}_{\text{w}}, \tag{1}$$

where $\bar{v}_{\text{w}} = M_{\text{w}}/\rho_{\text{w}}$, or from

$$(\rho_{\text{solution}} V_{\text{droplet}} - \rho_{\text{particle}} V_{\text{particle}})/M_{\text{w}}. \tag{2}$$

The calculated values of $SS_c$ were very close for the two approaches to calculation of droplet density, with a relative difference of only 0.1 % for dry particles with diameter of 40 nm, and less than 0.1 % for particles with a dry diameter of 100 nm. We therefore consider the assumption of volume additivity to affect the relevant models in a consistent and fairly even manner and not to impact the overall conclusions or interpretation of the results.

The sentence in question (Sect. 2.2.1 of the revised manuscript) has been clarified:

**The position of the Gibbs diving surface is selected so that the droplet bulk-phase volume equals the total equimolar droplet volume, and mass conservation ($n_j^{\text{T}} = n_j^{\text{S}} + n_j^{\text{B}}$) is assumed for all components in the droplet (water, surfactant, NaCl).**

We again emphasize that our purpose is not to evaluate or validate the underlying assumptions of the different models of cloud droplet surface activity. Our aim is to compare the predictions of the different bulk–surface partitioning models for the same systems and conditions, to assess the robustness of the current understanding in the community of the effects of aerosol surface activity on hygroscopic growth and CCN activation.

**1.3.2**

*Line 262. it seems like the CMC will depend rather strongly on the NaCl content. Please comment on this – can the uncertainty here be constrained?*

We have briefly mentioned the effects of NaCl on the CMC of the surfactant in Section 2.8 of the preprint. To account for this effect, it is necessary to use complex models with parameters obtained from experiments that are scarce or nonexistent for most relevant ternary systems. As explained above, our main purpose is not to evaluate the surface activity models used in the present work, or to provide constraints on the models or model parameters. Instead, our aim is to compare the predictions of the different bulk–surface partitioning models for the same droplet systems and conditions, to assess their robustness in capturing the properties of a given droplet system comprising surface active organic aerosol components.

We have expanded the discussion in Section 2.5 of the revised manuscript to further clarify the effects of NaCl on the surfactant CMC:

**In ternary water–surfactant–inorganic salt solutions, both the surfactant CMC and solution surface tension at the CMC can vary with inorganic salt concentration and may be further affected for ionic surfactants that share a common counterion with the inorganic salt. Accounting for these effects requires complex modeling (e.g., Kralchevsky et al., 1999; Calderón et al., 2020) with parameters obtained from experiments. The surfactant CMC often decreases with increasing inorganic salt concentration, compared to a binary water–surfactant solution (Calderón and Prisle, 2021).**

**However, for most ternary surfactant solutions, and in particular for atmospheric surfactants, the relevant interaction parameters are not known or readily accessible with existing experimental techniques.**

The discussion of the results around line 262 of the preprint (line 233 of the revised manuscript) has also been edited:

**However, the critical points for the Gibbs model correspond to droplet sizes immediately after the droplet surface tension increases from the minimum value at the $NaC_{14}$ CMC (see Fig. 2), suggesting that predictions could be sensitive to the assumed value of the CMC. Experimentally determined CMC values can vary significantly for a given solution, depending on the measurement technique used (Álvarez Silva et al., 2010).**

**1.3.3**

*Line 272-273. even if these acids were to partition very strongly, would the surface tension be suppressed? Does the mixture's surface tension depend on surface concentration necessarily? For example, pure liquids also can have a suppressed surface tension relative to water.*

The question pertains to the fundamental reason why bulk–surface partitioning models are needed. Detailed explanations for this point are given in the introductions of Lin et al. (2020) and Prisle (2021). We briefly highlight the main points here, together with clarifications made in the revised manuscript. The surface tensions of many pure, potentially sub-cooled, organic acids can be significantly lower than that of pure water (e.g. Hyvärinen et al., 2006; Vanhanen et al., 2008). This is the basis for the ability of these compounds to lower the surface tension of aqueous mixtures. When these compounds have enhanced surface concentrations, compared to an isotropic mixture, due to surface activity, the ability to reduce the surface tension of the aqueous solution is further enhanced. Bulk–surface partitioning in finite-sized droplets alters the surface enhancement at a given total concentration of the surface active solute, compared to a macroscopic solution, due to the simultaneous bulk-phase depletion. This effect is taken into account with the bulk–surface partitioning model. The model captures the bulk–surface partitioning at a given droplet state, and therefore the surface enhancement of the organic and resulting surface tension depression, as a function of either the bulk or surface concentration. Because experimental surface tension relations are far more readily obtained in terms of bulk concentration (e.g. Prisle et al., 2008, 2010; Malila and Prisle, 2018; Lin et al., 2018, 2020; Bzdek et al., 2020; Prisle, 2021), we typically use these relations to describe the surface composition and surface tension of the droplets. Surface tension is typically a decreasing function of surfactant bulk phase concentration. In microscopic droplets that contain a finite amount of solute, including the surface active compound, the surface tension depression may be dampened when the droplet bulk phase is strongly depleted from organic bulk–surface partitioning (Prisle et al., 2010; Bzdek et al., 2020; Prisle, 2021). In these droplets, also the surface concentration is reduced, compared to a macroscopic solution.

We have revised the Introduction of the revised manuscript to explain surface activity and related phenomena in more detail:

[revised manuscript text omitted]

**1.3.4**

*Line 445. can the authors make an estimate of the change in uncertainty in, e.g., cloud droplet number concentration, with the change in surface tension model? This is missing from the discussion.*

The different models that we use are not simply different surface tension models, but rather bulk–surface partitioning models which allow estimations of significantly altered bulk-phase concentrations for microscopic droplets, due to partitioning. These partitioning models are different, but together represent the current state-of-the-art of bulk–surface partitioning models available in the literature. In particular, the partitioning models employ various methods or assumptions to determine the droplet surface tension. The bulk solution model is included to the comparison as a reference that accounts for reduced surface tension, but does not take the effects of bulk–surface partitioning into account in estimating the surface tension of droplets. A more detailed explanation of why partitioning models are needed is provided above in Section 1.3.3 of this response.

We have added new results (Section 3.4 of the revised manuscript) showing the relative change in cloud droplet number concentration by considering organic aerosol surface activity with the different bulk–surface partitioning models and the bulk solution model, compared to a classical Köhler model which does not explicitly consider any effects of surface activity. The particle size range for these additional results has been extended to $D_\mathrm{p} = 50 - 200$ nm following the comment by reviewer 3 below in Section 2.5 of this response. Classical Köhler models are by far the most frequently used in larger-scale simulations, with only a few exceptions (e.g., Prisle et al., 2012a; Lowe et al., 2019). For completeness, we have now also included predictions with the classical Köhler model in the Results and Discussion and a short description of the classical Köhler model has been added in Section 2.4 of the revised manuscript:

**In the classical Köhler model (e.g. Prisle et al., 2010; Prisle, 2021), surface active aerosol components are treated as regular soluble solutes and specific effects of surface activity are not considered. The droplet surface tension is assumed to be constant and equal to that of pure water.**

The following explanation about estimating the relative change in the cloud droplet number concentration was added as a new Section 2.6 of the revised manuscript:

**We estimate the relative change in cloud droplet number concentration caused by the differences in critical super-saturation ($\mathrm{SS_c}$) predicted for the different partitioning models, as well as the bulk solution model, with respect to the classical Köhler model ($\mathrm{SS_c^0}$), using the method outlined by Bzdek et al. (2020). The cloud droplet number concentration $N$ is assumed to depend on the supersaturation SS as $N \propto \mathrm{SS}^k$, where $k \approx 0.5$ (Facchini et al., 1999), and the relative change in cloud droplet number concentration is calculated as**

$$\frac{\Delta N}{N} = \frac{(\mathrm{SS_c^0})^k - (\mathrm{SS_c})^k}{(\mathrm{SS_c})^k}. \tag{3}$$

190    The new results showing the relative change in the cloud droplet number concentration have been added as Section 3.4 of the revised manuscript:

[revised manuscript text omitted]

The Conclusions around line 445 of the preprint (lines 436 to 447 of the revised manuscript) have been edited to reflect these new additional results:

The different models can predict significantly different droplet activation properties ($\mathrm{SS}_c$, $d_c$, and $\sigma_c$) for the same strongly surface active particles. Differences between the predictions of the bulk–surface partitioning models increase with the surfactant mass fraction in the particles. Each partitioning model predicts or assumes strong partitioning of surfactant to the droplet surface, leading to a small overall Raoult effect and small variations in the droplet bulk water activity between the different partitioning model predictions. Predicted differences in the critical droplet properties for the investigated strongly surface active aerosols mainly stem from the droplet surface tension and ensuing Kelvin effect. Our results further show that cloud droplet number concentrations predicted for strongly surface active aerosol can vary significantly between the different models. Relative changes in cloud droplet number concentrations with respect to a classical Köhler model range from -35 % to 29 % between the five partitioning models, corresponding to both considerable warming and cooling climate effects. These differences represent a significant uncertainty in estimating the cloud radiative effects for regions where strongly surface active aerosol are prevalent. Therefore, conclusions regarding aerosol–cloud–climate effects of aerosol surface activity and bulk-surface partitioning based on any one surface activity model should not immediately be generalized.

**1.4 Technical corrections**

*Line 3. depleting the droplet bulk? This is a little ambiguous*

Clarification added:

...depleting the droplet bulk of the surfactant.

*Line 83. the notation of NaC14 is very confusing, as it seems to imply NaCl_4. Maybe the "14" could be a subscript?*

NaC14 has been changed to NaC$_{14}$.

*Line 90. extra period after "Table 1"*

260   The extra period has been deleted.

*Line 210. Extra period between sentence and citation*

The period has been moved after the citation.

*Line 261-262. this does not seem like a numerical artifact – I think the sentence needs to be clarified*

The sentence has been clarified:

265   **However, the critical points for the Gibbs model correspond to droplet sizes immediately after the droplet surface tension increases from the minimum value at the NaC$_{14}$ CMC (see Fig. 2), suggesting that predictions could be sensitive to the assumed value of the CMC.**

**2   Referee 3**

**2.1**

270   *This paper is written largely as a repetition of Part 1: S. Vepsäläinen et al.: "Comparison of six approaches to predicting droplet activation of surface active aerosol – Part 1: moderately surface active organics". The only differences I see are (1) that the six approaches are more completely described in Part 1, and (2) Parts 1 and the submission treat different aerosol: NaCl/Sodium myristate (NaC14) in the submission versus (NH4)2 SO4/Malonic acid in Part 1.*

Our overall purpose with these works is to compare Köhler predictions of cloud droplet activation with different bulk–
275   surface partitioning models for common systems and conditions, to assess the robustness of these-state-of-the-art frameworks currently in use by the community. This inter-model comparison should be carried out for a range of different types of surface active aerosols, for which the relative behavior of these models, as well as the behavior of each individual model, may differ significantly, due to variations in the expression of surface active properties. In Vepsäläinen et al. (2022), we compared surface activity models for common moderately surface active aerosol components, represented by malonic, succinic, and glutaric
280   acids. We focused the discussion on the results for malonic acid, due to the similarity between the results for these three organic acids, whereas results for succinic and glutaric acid aerosols are presented in the supplement of Vepsäläinen et al. (2022). However, it is not given that the inter-comparison between the different surface activity models can immediately be translated to other surface active aerosol systems or conditions, where the effects of surface activity on cloud droplet activation may be expressed differently. The surface adsorption properties differ between strong surfactants and moderately surface active
285   compounds. In this work, we therefore investigate whether the results of the inter-comparison between the different surface activity models in Köhler theory for common systems of moderate surface active compounds also apply for the selected strongly surface active aerosols.

We find both similarities and significant differences between the partitioning model inter-comparison for the strongly surface active aerosols studied in the present work and the moderately surface active compounds previously studied by Vepsäläinen

et al. (2022). Therefore, the robustness of the partitioning model predictions cannot be assessed from comparing predictions for only a single or few surface active aerosol systems. Conclusions made for any one type of aerosol cannot be generalized for different surface active aerosols without investigating a variety of systems.

**2.2**

*For example Part 1 provides a conceptual figure of the different models that is not present in the submission. To get such detail the reader has to go back and forth between the two papers (Part 1 and submission ) anyway – so why not, in the submitted paper, save journal space and simply refer to Part 1 for the six models and the theory already described there throughout?*

We have carefully considered how to strike the balance between the detailed descriptions included in Vepsäläinen et al. (2022) and providing necessary information readily available for the present work with a minimum of repetition. We prefer to provide relevant information on the key concepts in the present manuscript. In response to the reviewer's comments about saving journal space, we have attempted to further condense the descriptions in the current work. We have added a new table (Table 2 of revised manuscript) that presents a summary of the different models, allowing for somewhat shorter model descriptions. However, there are differences between the current manuscript and Vepsäläinen et al. (2022) that make further condensation of the model descriptions difficult. In particular, the explanation of relevant details, such as the model-specific parameters and parameterizations for the physicochemical properties of the droplet solution, etc., require sufficient context. The most significantly shortened descriptions are the compressed film model (Section 2.2.3 of the revised manuscript), and the partial organic film model (Section 2.2.5 of the revised manuscript). The other model descriptions have only been slightly edited.

During the revisions, we also added a classical Köhler model to the comparison, to elaborate on potential larger-scale atmospheric effects that could be caused by the differences between the predictions of the different surface activity models. The details can be found above in Section 1.3.4 of this response. The classical Köhler model is briefly described together with the other models in Section 2.4 of the revised manuscript.

**2.3**

*The authors refer frequently to Part 1, but I find these comparisons to be more descriptive than insightful. It would be useful to provide the reader with a broader understanding/overview of the effects of moderate versus strong surfactants learned from the two papers as a whole. Part 1 succeeds better in this respect as well, with a conclusions section that makes broad connections of that work to cloud microphysics, fluctuations in supersaturation, and other large-scale effects. The submitted manuscript gives neither discussion nor conclusions, which is suprising in that there are now two papers in the series to draw from.*

Our purpose with both the present work and previous work of Vepsäläinen et al. (2022) is not to assess or compare the effects of strong and moderately surface active aerosols in cloud droplet activation. We are comparing the predictions of different bulk–surface partitioning models for the same aerosol systems and conditions, in order to assess the robustness of these existing state-of-the-art models currently in use, for different types of surface active aerosol systems. We perform this inter-model comparison for different types of surface active aerosols, where the expression of surface activity and the relative ability of the different surface activity models to capture these properties may differ. The robustness of our understanding of the

comprehensive effects of surface active aerosol on cloud droplet activation that is revealed by the comparison of these existing models is investigated for 1) the specific surface active aerosol systems treated in each of the manuscripts, and 2) between aerosol systems with different surface active properties. We have further emphasized and clarified these points and the purpose of this work in the Results and Discussion, as well as the Conclusions of the revised manuscript. Additionally, we have added subsections to the Results and Discussion of the Köhler curves for a clearer presentation of the results. Some of the purely descriptive comparisons between the predictions for strongly and moderately surface active aerosol have been removed in the revised manuscript.

To elaborate on the connections of the present partitioning inter-model comparison to larger-scale phenomena, we have added new results and discussion (Section 3.4 of the revised manuscript) showing the relative change in cloud droplet number concentration from Köhler predictions with the different partitioning models and the bulk solution model. The relative change is calculated with respect to a classical Köhler model that does not include effects of surface activity. The Conclusions have also been edited to reflect these new results. More details have been given above in response to the specific comment from reviewer 2 in Section 1.3.4 of this response. The discussion related to the calculated Köhler curves has been extended in the Results and Discussion Section 3.1.1 of the revised manuscript (starting from line 262):

**Average supersaturations in low-level clouds range from 0.1 % to 0.4 % (e.g. Politovich and Cooper, 1988) but higher supersaturations of 0.7 % to 1.3 % can be reached during turbulent fluctuations of temperature and water vapor concentration (e.g. Siebert and Shaw, 2017). Only the simple partitioning, compressed film, and classical Köhler models predict $SS_c$ above 0.7 % at high $w_{p,sft}$, while the other models predict considerably lower $SS_c$ (Table 3). Therefore, the inter-model variation for the strongly surface active aerosol studied here is sufficiently pronounced to affect predictions of activation for the ambient conditions in low level clouds. The absolute differences between the highest and lowest $SS_c$ in Fig. 1 are $\Delta SS_c = 0.27, 0.41, 0.69$, and $1.30$ % for $w_{p,sft} = 0.2, 0.5, 0.8$, and 0.95, respectively. However, excluding the bulk solution model, maximum differences in $SS_c$ between the bulk–surface partitioning models are $\Delta SS_c = 0.10, 0.24, 0.54$, and $1.28$ % for $w_{p,sft} = 0.2, 0.5, 0.8$, and 0.95, respectively. This range of $\Delta SS_c$ for strongly surface active $NaC_{14}$ particles is similar to predictions for malonic acid by Vepsäläinen et al. (2022). The large differences between the $SS_c$ of the different models for both strongly and moderately surface active aerosol means that representation of surface activity during Köhler calculations could cause significant changes in predictions of cloud droplet number concentrations (Fig. 4), with corresponding uncertainty in estimations of the cloud radiative effect.**

To expand the discussion about surface tension, we have added the following at the end of Section 3.2 of the revised manuscript (starting from line 352 of the revised manuscript):

**Droplet surface tension impacts the conditions for growth and activation via the Kelvin term of the Köhler equation (1). The importance of surface tension in cloud droplet formation has been a topic of debate for decades (e.g. Li et al., 1998; Sorjamaa et al., 2004; Booth et al., 2009; Prisle et al., 2008, 2010; Nozière et al., 2014; Gérard et al., 2016; Ruehl et al., 2016; Ovadnevaite et al., 2017; Davies et al., 2019; Lowe et al., 2019; Bzdek et al., 2020). Here, we observe that different models predict diverging droplet surface tensions for strongly surface active aerosol, similarly to predictions for moderately surface active aerosol by Vepsäläinen et al. (2022). The $\sigma_c$ of the monolayer and bulk solution models for**

**strongly surface active aerosol are significantly lower than for moderately surface active aerosol, while the other partitioning models predict similar $\sigma_c$ for both strongly and moderately surface active aerosol. The predicted droplet surface tensions reflect the underlying assumptions of the different models (Table 2), which do not account for all surfactant properties to the same degree. This suggests that varying conclusions about the importance of surface tension in cloud droplet activation could be partly due to differences between the various surface activity models applied. These model differences show that caution should be taken when interpreting the role of surface tension based on the predictions of any given model.**

We have edited the discussion in Section 3.3 (starting from line 375 of the revised manuscript) of the revised manuscript to focus more on potential effects for aqueous chemistry:

**The strong partitioning predicted (or assumed) with the different partitioning models for $NaC_{14}$ means that inter-model differences in predicted droplet activation in Fig. 1 are mainly caused by differences in the droplet surface tension (Fig. 2). Surface activity could also have important implications for a variety of processes related to cloud microphysics, including aqueous droplet chemistry (Prisle, 2021). Chemical reactions in aqueous aerosols can be accelerated relative to macroscopic solutions (Marsh et al., 2019). Strong partitioning of surface-active species and the simultaneous depletion of the droplet bulk phase can affect the chemical environment in the droplet, in particular in the submicron range (Prisle et al., 2010), changing possible reaction pathways and rates in the surface (Prisle et al., 2012b; Öhrwall et al., 2015; Werner et al., 2018) and bulk (Prisle, 2021) phases of the droplet. Interfacial reactivity could be pronounced due to the large surface-area-to-volume ratios of finite volume droplets (Prisle et al., 2012b; Bzdek et al., 2020; Prisle, 2021).**

We have also added more emphasis on the inter-model comparison between the strongly and moderately surface active aerosols studied between the present manuscript and Vepsäläinen et al. (2022), both in the Results and Discussion and Conclusions of the revised manuscript. In the Results and Discussion, we have added the following as Section 3.1.3:

**For particles containing strong surfactant $NaC_{14}$ with NaCl in Fig. 1, the simple partitioning and compressed film models predict the highest $SS_c$ at high surfactant fractions $w_{p,sft}$. The monolayer and partial film models agree well for the entire range of particle compositions, while the Gibbs model predicts comparable $SS_c$. In previous Köhler calculations for particles containing moderately surface active malonic, succinic, or glutaric acid with ammonium sulphate (Vepsäläinen et al., 2022), the highest $SS_c$ were predicted with the simple partitioning model, while the Gibbs, monolayer, and bulk solution models agreed well, and predicted similar $SS_c$ to the compressed film and partial film models for most $w_{p,sft}$. The inter-comparison of surface activity and partitioning models therefore show several differences between strongly and moderately surface active aerosols. This suggests that effects of aerosol surface activity on cloud droplet formation captured by the different partitioning models cannot be robustly understood based on any single model or a few aerosol systems with similar surface activity. Therefore, assessments of bulk–surface partitioning effects in Köhler calculations should be based on predictions for a wide range of conditions and types of surface active aerosol relevant to the atmosphere.**

More emphasis on the inter-model comparison between different aerosol systems has been added also in the Conclusions (starting from line 448 of the revised manuscript):

**Comparison of inter-model variation for strongly and moderately (Vepsäläinen et al., 2022) surface active aerosol shows how the mutual agreement between the different surface activity and partitioning models varies with aerosol surface activity. Therefore, conclusions about the robustness of our understanding of the effects of aerosol surface activity, as captured by the inter-model variation, do not immediately translate between different surface active aerosol systems. This emphasizes the need to validate aerosol surface activity models for a range of surface active aerosol types and ambient conditions, before establishing their broad applicability in atmospheric modeling. Generalization of Köhler predictions for only a few surface active aerosol systems and conditions could introduce significant bias in modeling larger-scale atmospheric processes.**

**2.4**

*In summary, the authors should shorten their paper given the repetitive overlap with Part 1 noted in the first paragraph of this review. Moreover, the authors should end their paper by providing general discussion along the lines described in the second paragraph of this review – if for no other reason than to contextualize their point-by-point and figure-by-figure comparisons made throughout the paper.*

The reviewers' comments concerning 1) relation of the current manuscript to Vepsäläinen et al. (2022) and 2) shortening of the manuscript are both discussed above in Sections 2.1 and 2.2 of this response. The Results and Discussion of the revised manuscript now also 3) contains a broader discussion about the potential larger-scale effects caused by differences between the predictions of the various models for aerosols containing $NaC_{14}$. Furthermore, 4) more focus has been given to clarify the importance of the inter-model comparison between the strongly and moderately surface active aerosols studied between the current work and Vepsäläinen et al. (2022). The details relating to points 3) and 4) can be found in Section 2.3 of this response.

**2.5**

*Finally, the authors should consider, perhaps in a separate figure, expanding the range of particle size, now limited to the single size of Dp = 50nm. The most important location along any Kohler curve is the critical point, or maximum, marking the threshold for cloud droplet activation. Showing a locus of these points as particle size is varied for one or more of the six models would add this new dimension.*

We have extended the calculations for a wider particle size range of $D_p = 50 - 200$ nm. New results are presented in Section 3.4, where we present the relative change in the cloud droplet number concentration in response to feedback from reviewer 2 (Section 1.3.4 of this response). We have also included additional results for the same extended dry particle size range in the supplementary material and added the following at the beginning of the Results section:

[revised manuscript text omitted]